# Turning Uncertainty into Control: Bi-level Training with Editable Bayesian Layers

## Abstract

As deep learning systems are increasingly deployed in high-stakes settings, it is essential not only to quantify predictive uncertainty but also to use it to steer training and improve downstream decision. Yet dense parameter updates often cause collateral interference due to the entangled nature of parameter. In this paper, We propose **B**ayesian **F**eature **R**eweighting (**BFR**), a framework that turns calibrated, instance-level uncertainty into training-time control signals while enforcing sparse, localized parameter updates. Concretely, BFR employs a Bayesian Last Layer to obtain well-calibrated predictive uncertainty with negligible overhead and imposes sparsity on the decision layer to concentrate changes on task-relevant parameters, thereby limiting unintended perturbations. It then tinkers the model via uncertainty-guide bi-level optimization. Across vision and language benchmarks, BFR consistently improves group robustness while maintaining competitive accuracy and yields smaller parameter changes. Code is provided in the supplementary material.

## 1 Introduction

Deep learning has witnessed remarkable progress over the past decades across a wide spectrum of tasks (LeCun et al., 2015). While it has achieved notable success in scientific and industrial applications, enhancing model reliability and robustness remains a pressing challenge in safety-critical, high-stakes domains such as medical diagnosis, autonomous driving, and financial risk management (Amodei et al., 2016; Senge et al., 2014). In these scenarios, uncertainty estimation has emerged as a key mechanism for constructing trustworthy learning systems, owing to its ability to provide calibrated confidence scores; its primary aim is to ensure that neural networks assign low confidence to test cases that are poorly represented by the training data or prior knowledge (Gal & Ghahramani, 2016; Lakshminarayanan et al., 2017; Fan et al., 2021), enabling risk-aware decision making.

In recent years, driven by growing application demands and pratical deployment, research on uncertainty estimation has broadened from how to obtain uncertainty to how to use it effectively and reliably. On the one hand, lightweight modeling and inference seek to deliver stable confidence estimates and good calibration under tight computational and memory budgets (Daxberger et al., 2021; Harrison et al., 2024); on the others, work targeting robustness and interpretability introduces structural modeling and explicit constraints to enhance the usability and auditability of uncertainty representations (Wang et al., 2025; Hu et al., 2025). Despite these advances in improving the credibility of uncertainty itself, a growing line of work has begun to apply uncertainty estimation to real-world tasks (Geifman & El-Yaniv, 2019; Kendall et al., 2018); Nevertheless, developing uncertainty-guided mechanisms that systematically use predictive uncertainty as a control signal during training, to enhance representation learning and decision quality, remains an open challenge.

To enable uncertainty estimation to genuinely guide representation learning, one not only provides calibrated signals but also enacts parameter-efficient controlled interventions, specifically, editable updates that target minimal parameters while leveraging scarce supervision (Geva et al., 2020). Such interventions should prioritize localized parameter adjustments, modifying only a sparse subset of the model's weights to avoid disrupting well-learned representations. A central difficulty is that parameters in deep neural networks are highly entangled, so dense updates often induce unintended behavioral drift and collateral interference (McCloskey & Cohen, 1989; Kirkpatrick et al., 2017). To mitigate this, recent work leverages structured sparsity (Frantar & Alistarh, 2023; Wang et al., 2024)

to increase the locality and selectivity of parameter updates. These studies indicate that sparsity constraints help concentrate changes on the subset of parameters most relevant to the target knowledge, thereby minimizing perturbations to the rest of the model.

Considering the inherent limitations discussed above, the Bayesian Last Layer (BLL) technique (Weber et al., 2018) emerges as a compelling solution: t offers a pathway to well-calibrated uncertainty estimates with only modest additional probabilistic training cost, while naturally accommodating parameter priors—enabling effective sparsification without compromising accuracy (Hu et al., 2025). While prior work has mainly used BLL as a post-hoc, inference-time uncertainty estimator, here we integrate it directly into the training process and use its posterior as a principled control signal for uncertainty-guided feature learning.

To make this concrete, we instantiate it under a spurious correlation setting, which refers to misleading patterns that appear in the training data but fail to generalize to unseen environments. In this setting, samples are often partitionable into groups by environment or attribute, and robustness is assessed via the worst-group metric; those governed by non-causal cues frequently form latent minority subgroups whose group labels are typically unavailable during training. (Michel et al., 2022; Sagawa et al., 2020) Unlike class labels that only specify the semantic category (e.g., "cat" vs. "dog"), group annotations encode subpopulation structure such as class–background or class–attribute combinations (e.g., "dog on grass" vs. "dog indoors"), and thus typically require additional domain knowledge or metadata, making reliable group labels costly and often unavailable in practice (Michel et al., 2022; Sagawa et al., 2020). Recent work therefore seeks to improve subpopulation robustness with reduced group supervision (Qiao et al., 2025; To et al., 2025; Li et al., 2024), but most methods still rely on a small number of group labels for model selection or repeated full-model retraining. In this context, uncertainty estimation offers a natural training-time control signal: it empirically correlates with rare, error-prone subgroups and can be injected into a differentiable objective to prioritize such samples without any explicit group annotation.

However, in this setting, how to effectively leverage uncertainty remains an open problem. Directly weighting training samples by predictive uncertainty often leads to convergence issues (Kendall & Gal, 2017). High-uncertainty samples can receive unbounded weights, causing gradient explosions, whereas low-uncertainty samples are nearly discarded, yielding sparse gradients that optimizers struggle to balance (Seitzer et al., 2022). To circumvent this, we avoid directly reweighting samples by uncertainty. Instead, we employ a bi-level optimization scheme (Qiao et al., 2025; Ren et al., 2018) that updates weights dynamically. Specifically, we split the validation set into a low-uncertainty support set and a high-uncertainty target set via instance-level predictive uncertainty. A meta-update minimizes support loss while maximizing a lower bound on the target objective, stabilizing training and emphasizing implicit minority subgroups without group annotations.

Combining these ideas, we propose **B**ayesian **F**eature **R**eweighting (**BFR**), which turns uncertainty into a training-time control signal without group labels, and enforces localized parameter updates via a sparse constrain: **(1) Probablistic Pretraining** We reinterpret the softmax classification as a generation process for labels and replace the conventional linear classifier with BLL. This enables a probabilistic formulation of the network, modeling the feature and classifier weights as stochastic latent variables with sparsity-inducing priors, thereby capturing both aleatoric and epistemic uncertainty; **(2) Uncertainty-guided Bi-level Optimization**. During the second phase, we obtain an instance-level predictive uncertainty captured by $\theta$ and $\Phi$. Based on this signal, we partition the data into *(i)* a support set of low-uncertainty examples and *(ii)* a target set of high-uncertainty examples. We then train the BLL with a bi-level objective. This learn-to-reweight procedure dynamically adjusts uncertainties, amplifying the decision weight on implicit minority subpopulations.

Our contributions can be summarized as follows:

- We introduce **B**ayesian **F**eature **R**eweighting (**BFR**), a general framework for in-training utilization of uncertainty and performs parameter updates with controlled impact.

- To ensure stability in uncertainty-driven training process under group-annotation-free settings, we adopt a bi-level optimization scheme that leverages uncertainty to dynamically partition data and adaptively adjust sample weights.

- Extensive experiments across diverse architectures and data modalities demonstrate that BFR effectively improves group robustness and generalization without group annotations.

## 2 RELATED WORKS

### 2.1 UNCERTAINTY ESTIMATION

Work on predictive uncertainty goes beyond point estimates. Ensembles aggregate stochastic networks (Lakshminarayanan et al., 2017; Liu et al., 2021b); Bayesian NNs put priors on all weights but are hard to train (Blundell et al., 2015; Hernández-Lobato & Adams, 2015); Bayesian Last Layers (BLLs) make only the head probabilistic, enabling lightweight retraining/VI (Weber et al., 2018; Harrison et al., 2023; Watson et al., 2021; Hu et al., 2025); Evidential DL models higher-order (e.g., Dirichlet) evidence (Sensoy et al., 2018; Malinin & Gales, 2018; Malinin et al., 2020). Beyond estimation, a growing body of work leverages uncertainty during training for downstream objectives. For selective prediction under risk–coverage (Geifman & El-Yaniv, 2019) and uncertainty-guided sample reweighting (Chang et al., 2017; Rizve et al., 2021). In contrast, **BFR** integrates calibrated instance-level uncertainty as a control signal and employs sparsity constraints to intervene in a controlled, localized manner, enabling editable parameter updates while limiting collateral perturbations.

### 2.2 SPURIOUS CORRELATION MITIGATION

In spurious-correlation settings, empirical risk minimization (**ERM**)(Vapnik, 1999) minimizes average loss, boosting mean accuracy but ignoring subgroup heterogeneity. Group-robust methods optimize worst-group accuracy yet typically require subgroup annotations (Sagawa et al., 2019). Truly group-annotation-free regimes—no group annotations in train and val-are rarer, though recent work narrows the gap via pseudo groups from VLMs (Zheng et al., 2024), clustering-derived subgroups (Sohoni et al., 2020; To et al., 2025), saliency-based spuriosity detectors (You et al., 2025; He et al., 2025), or per-example reweighting with full retraining (Li et al., 2024). In contrast, **BFR** uses instance-level uncertainty—without auxiliary group generators or full-model retraining—to improve group robustness.

## 3 BAYESIAN FEATURE REWEIGHTING

In this section we present ***Bayesian Feature Reweighting*** (***BFR***), whose overall workflow is depicted in Fig. 1. In the first stage, BFR replaces the deterministic classifier head with a Bayesian Last Layer (BLL) endowed with sparsity-inducing priors and performs ***Probabilistic Pretraining***, thereby explicitly modeling both aleatoric and epistemic uncertainty while enforcing sparsity to localize decision making. in the second stage, BFR refines the model via ***Uncertainty-guided Bi-level Optimization***, which emphasize informative high-uncertainty samples in a stable, feedback-driven manner. Before delving into the details, we first present the necessary preliminaries.

### 3.1 PRELIMINARIES

Considering a classification problem, the typical approach is Empirical Risk Minimization (ERM), which minimizes the expected training loss:

$$L_{\text{ERM}}(f_\omega) = \min_{\omega \in \Omega} E_{(x,y) \sim D_{\text{train}}} \left[ \ell\big(f_\omega(x),\, y\big) \right], \tag{1}$$

in which $f_\omega$ is a learned neural network mapping $\mathcal{X}$ to $\mathcal{Y}$, parametrized by $\omega$ from the parameter space $\Omega$. $\ell$ is the loss function. We now adopt a latent variable model to re-examine the ERMs, where $f_\omega$ are commonly tackled by training domain-specific neural networks with a sigmoid or softmax output layer, which can be formulated as:

$$p(y \mid x) = \frac{\exp\{z w_y^T + b_y\}}{\sum_{y' \in \mathcal{Y}} \exp\{z w_{y'}^T + b_{y'}\}}, \tag{2}$$

$z$ is the representation of ERM, $w_y$ and $b_y$ denote the weight and bias of the last fully-connected layer. Previous work (Joo et al., 2020) shows that softmax classifier is a special case of Eq. 3:

$$p(y \mid x) = \int_z p(y \mid z) p(z \mid x), \tag{3}$$

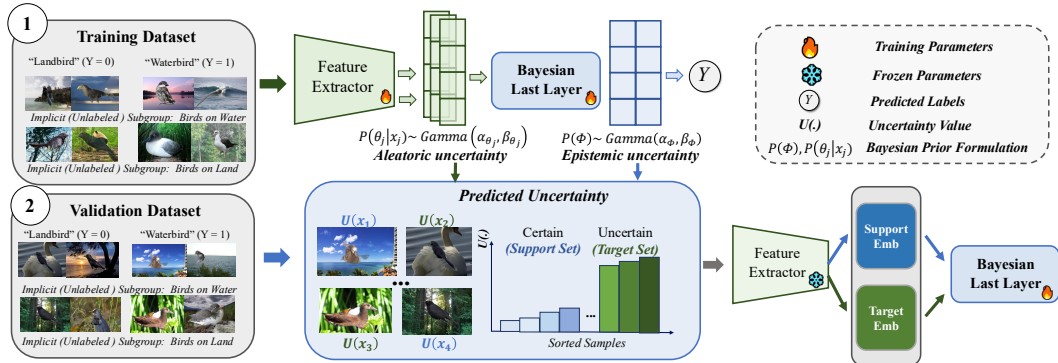

Figure 1: Overall framework of BFR. (1) **Probabilistic Pretraining**: We replace the deterministic classifier with a designed BLL; treats the feature $z$ and class weights $w_y$ as stochastic latent variables $(\theta, \Phi)$ and imposes the sparse, non-negative priors on them to obtain disentangled parameters. (2) **Uncertainty-guided Bi-level Optimization**: After pretraining, we rank validation examples by predictive uncertainty to form a high-uncertainty target set and low-uncertainty support set. A bi-level optimization to the BLL then minimizes support loss while adaptively optimizing on the target, preserving stable features and amplifying on implicit minority subgroups.

$f_\omega$ treats $p(z \mid x)$ as a point mass at $z = f(x)$; the softmax then sets $p(y \mid z)$. Thus, $y$ can be viewed as a sample from $p(y \mid x)$, *i.e.*, $y \sim p(y \mid x)$. However, due to the absence of stochasticity in the latent variable $z$, $p(y \mid x)$ remains a deterministic mapping and is thus often poorly calibrated, leading to over-confident predictions that capture model uncertainty inadequately(Guo et al., 2017).

## 3.2 PROBABILISTIC PRETRAINING

To obtain better calibration and faithfully characterize uncertainty, we adopt the Bayesian Last Layer (Hu et al., 2025) paradigm to reformulate the network. Concretely, We recast Eq. 3 as a fully probabilistic model by treating the feature $z$ and the class-specific weights $w_y$ as stochastic latent variables: $z$ becomes the local latent $\theta$, and $w_y$ the global latent $\Phi$. The revised formulation is:

$$p(y \mid x) = \int_{\theta, \Phi} p(y \mid \theta, \Phi) p(\theta \mid x) p(\Phi), \tag{4}$$

and the corresponding generative model is defined as:

$$y_j \mid \theta_j \sim \text{Category}(\theta_j \Phi), \quad \theta_j \mid x_j \sim \text{Gamma}(1, 1), \quad \Phi \sim \text{Gamma}(1, 1). \tag{5}$$

Both $\theta$ and $\Phi$ are assigned $Gamma(1, 1)$ prior, which enforces sparsity and non-negativity, promoting more disentangled feature learning and improving model identifiability (Zhou et al., 2012). This formulation mirrors the nature of aleatoric and epistemic uncertainty (Kendall & Gal, 2017): the former reflects irreducible noise inherent in the data, whereas the latter captures uncertainty stemming from the model itself. Similar to VAEs (Kingma et al., 2015), the optimization objective can be achieved by maximizing the evidence lower bound (ELBO) of the log-likelihood as:

$$\mathcal{L}(Y) = \sum_{j=1}^{J} \mathbb{E}_{q(\theta_j|x_j)}[\ln p(\mathbf{y}_j \mid \theta_j, \Phi)] - \sum_{j=1}^{J} \mathbb{E}_{q(\theta_j|x_j)}\left[\ln \frac{q(\theta_j \mid x_j)}{p(\theta_j)}\right] - \mathbb{E}_{q(\Phi|-)}\left[\ln \frac{q(\Phi \mid -)}{p(\Phi)}\right], \tag{6}$$

where $J$ is the number of training samples and the first term is the expected log-likelihood of the generative model, which ensures reconstruction performance, and the last two term is the Kullback–Leibler (KL) divergence that constrains the variational posterior distribution $q(-)$ to be close to its prior $p(-)$. Eq. 6 follows the standard ELBO formulation and is instantiated here to generate the training-time uncertainty control signal used in Sec. 3.3–3.5.

## 3.3 UNCERTAINTY-GUIDED BI-LEVEL OPTIMIZATION

The probabilistic pretraining equips the model with uncertainty estimation and, under sparse, non-negative priors, induces sparser, more disentangled representations—laying the foundation for the

uncertainty-guided, localized bi-level optimization developed in this section. To enhance training stability, the parameters are updated indirectly through the meta-objective. Specifically, after probabilistic pre-training, the stochasitic latent variable $\theta$ and $\Phi$ capture aleatoric and epistemic uncertainty, respectively; their combination constitutes the model's predictive uncertainty (Mukhoti & Gal, 2018). Given a test input $x_j$ and the training set $D_{train}$, we first define the class-wise Monte-Carol-sampled probability in BFR as:

$$\mathbf{P}_{x_j} = [P_{c,t}]_{c=1,\ldots,C}^{t=1,\ldots,T} = [\, p(y_j = c \mid x_j, \hat{\theta}_j^{(t)}, \hat{\Phi}^{(t)}) \,]_{c,t}.$$

Here $\mathbf{P} \in \mathbb{R}^{C \times T}$, in which $C$ and $T$ is the number of classes and Monte Carol samples (stochastic forward passes), respectively. $p(y_j = c \mid .)$ is the softmax probability that input $x_j$ belongs to class $c$, where $\hat{\theta}_j^t$ and $\hat{\Phi}^t$ denote the model parameters from the $t$-th Monte Carlo sample. Then the class-wise average probability can be further defined as:

$$s_{c,j} \;=\; \frac{1}{T}\sum_{t=1}^{T}\mathbf{P} = \frac{1}{T}\sum_{t=1}^{T} p\Big(y_j = c \mid x_j, \hat{\theta}_j^t \hat{\Phi}^t\Big). \tag{7}$$

Unlike previous information theoretic metrics, we use a statistical test based method, *i.e.*, $p$-value, to estimate uncertainty. One advantage of using hypothesis testing over information theoretic metrics is that the $p$-value of the test can be more interpretable, making it easier to be deployed in practice to obtain a binary uncertainty decision (Papadopoulos, 2008). Hence, we use the two-sided test $p$-value as the instance-level uncertainty estimate to quantify how confident our model is about this prediction, but unlike contextual droupout (Fan et al., 2021), our measure is computed from posterior samples drawn from a Bayesian last layer, rather than from stochastic dropout activations. This distinction ensures that the resulting $p$-value reflects genuine epistemic–aleatoric uncertainty derived from a probabilistic model. Specifically, we evaluate whether the difference between the empirical distributions of the two most possible classes from multiple posterior samples is statistically significant. Let $s_{(1),x_j} \geq s_{(2),x_j} \geq \cdots \geq s_{(|\mathcal{C}|),x_j}$ denote the descending order of $\{s_{c,x_j}\}_{c\in\mathcal{C}}$; then we select the top-2 indices according to the two largest scores in the ordering:

$$c_{1,x_j}^{\star} = \arg\max_{c\in\mathcal{C}} s_c, \qquad c_{2,x_j}^{\star} = \arg\max_{c\in\mathcal{C}\setminus\{c_1^{\star}\}} s_c. \tag{8}$$

The two-sided $p$-value computed between $\mathbf{P}\big[c_{1,x_j}^{\star}\big]$ and $\mathbf{P}\big[c_{2,x_j}^{\star}\big]$ is taken as the uncertainty measure for the input sample, *i.e.*, $U(x_j) = p_{two-sided}$, which is defined as:

$$p_{\text{two-sided}} = 2\big[1 - F_{t_{T-1}}(|t_{\text{stat}}|)\big]$$

$$d_t = \mathbf{P}\big[c_{1,x_j}^{\star}, t\big] - \mathbf{P}\big[c_{2,x_j}^{\star}, t\big], \quad t = 1,\ldots,T. \quad t_{\text{stat}} = \frac{\bar{d}}{\sigma_d/\sqrt{T}}. \tag{9}$$

Here, $\bar{d}$ denotes the sample mean of the paired differences $d_t$ and $\sigma_d$ is the sample standard deviation. Under the null hypothesis of zero mean difference, the test statistic $t_{stat}$ follows a Student-$t$ distribution with $T-1$ degrees of freedom, whose cumulative distribution function is $F_{t_{T-1}}$. Higher value of $U(x)$ suggest larger predictive uncertainty for instance $x$; in practice, we declare examples as "uncertain" when the two-sided test fails to reject the null of equal top-two predictive means at the 5% level. The choice threshold equals to $0.05$ follows common statistical practice for error control rather than a domain-specific optimality claim; consistent with prior guidance (Wasserstein & Lazar, 2016; Angelopoulos & Bates, 2021), we treat it as a conventional decision threshold and additionally report **sensitivity to** $\{0.01, 0.05, 0.10\}$ **in Appendix D.3**.

Retraining the last layer solely on the high-uncertainty set typically causes convergence issues. Assigning excessively large weights to these samples enlarges gradient disparity relative to the rest, leading to optimization instability or even gradient explosion (Qiao & Peng, 2021; Kendall & Gal, 2017). Instead of using $p$-value as heuristic uncertainty scores, we use it as statistically interpretable indicators for constructing the support set and the target set under bi-level optimization scheme (Zhou et al., 2022; Shu et al., 2019). Formally, this uncertainty-driven partition can be interpreted as an uncertainty-conditioned decomposition of the data distribution,

$$p(X) = p(X \mid U(x) < \delta) + p(X \mid U(x) \geq \delta)$$

where $U(x)$ denotes the uncertainty estimator. This decomposition separates the empirical distribution into low- and high-uncertainty regions, corresponding to the support and target sets, respectively. To minimize the worst-group risk, the bi-level optimization can be formulated as a minmax problem (Zhou et al., 2022):

$$\min_{\mathbf{v}\in\mathcal{V}} \max_{g\in\mathcal{G}} \mathbb{E}_{(x,y)\in D_g^{tar}}[\ell(f_{\hat{\omega}^*}(x),y)] \quad s.t. \quad \hat{\omega}^* = \arg\min_{\omega} \mathbb{E}_{(x,y)\in D^{sup}}[\ell(f_\omega(x),y,\mathbf{v})] \quad (10)$$

where $\mathbf{v}\in\mathbb{R}^n$ be the weight vector for all training samples and $D^{tar}$ and $D^{sup}$ is the used target set and support set correspondingly. The inner loop fixes the sample weights $\mathbf{v}$ and optimizes $\omega$ on the support set $D^{sup}$ to obtain $\hat{\omega}^*$; the outer loop then updates $\mathbf{v}$ with $\hat{\omega}^*$ fixed, minimizing the worst-group risk on the target set $D^{tar}$ and thus focusing the model on minority groups. Since BFR dispenses with group annotations, we designate the high-uncertainty samples as the **target set** and treat the remaining low-uncertainty samples as the **support set**. With this uncertainty-based partition, the bi-level formulation in Eq.10 specializes to:

$$\min_{\mathbf{v}\in\mathcal{V}} \max_{\pi\in\Pi} \mathbb{E}_{(x,y)\in\pi}[\ell(\tilde{f}_{\hat{\omega}^*}(x),y)]$$
$$s.t. \quad \hat{\omega}^* = \arg\min_{\omega} \mathbb{E}_{(x,y)\in D^{sup}}[\ell(\tilde{f}_\omega(x),y,\mathbf{v})], \quad \tilde{f}(x,\omega)\sim r(\omega), \quad (11)$$

where $\tilde{f}(x;\omega)$ denotes a single stochastic forward pass for BLL-based models, $r(\omega)$ is the posterior distribution over the network parameters, $\Pi$ denotes a family of distributions supported on $D^{tar}$ and $\max_{\pi\in\Pi}$ takes the worst-case over the unknown implicit subgroups. The outer-inner optimization in Eq. 11 aligns with the empirical Distributionally Robust Risk Minimization framework (DRRM): $\min_\omega \max_{P\in\mathcal{U}(P_0)} \mathbb{E}*(x,y)\sim P[\ell(f*\omega(x),y)]$, where $\mathcal{U}(P_0)$ is an set centered at the empirical distribution $P_0$. Unlike conventional DRRM methods that require explicit group annotations to construct $\mathcal{U}(P_0)$, BFR induces implicit sub-groups from predictive uncertainty, achieving worst-group robustness without group annotations. To solve Eq. 11 stably, we employ distinct optimization strategies for the inner and outer optimization:

**(1) Inner-loop (last-layer retraining)**. With the sample weights $\mathbf{v}$ fixed, we update only the Bayesian last–layer parameters by minimizing the weighted loss on the support set $D^{sup}$, the weighted loss can be derived from Eq. 6 as:

$$\mathcal{L}_{in}(Y;\mathbf{v}) = \sum_{j=1}^{|D^{sup}|} v_j \mathbb{E}_{q^*(\theta_j^*|x_j)}[\ln p(\mathbf{y}_j\mid\theta_j,\Phi)] - \mathbb{E}_{q(\Phi|-)}\left[\ln\frac{q(\Phi\mid-)}{p(\Phi)}\right], \quad (12)$$

where $\theta$ is sampled from the trained posterior $q^*(\theta_j^*\mid x_j)$ of probabilistic pretraining stage, and we only optimize $\Phi$ in the inner loop.

**(2) Outer-loop (weight update)**. We compute the gradient with respect to $\mathbf{v}$ via implicit differentiation (Qiao et al., 2025; Krantz & Parks, 2002). Because the resulting gradient expression does not explicitly contain $\mathbf{v}$, the weights influence the update only through their effect on the inner-level optimum $\hat{\omega}^\star$, thereby mitigating gradient explosions caused by extreme weights. Intuitively, $\mathbf{v}$ is adjusted according to the impact of the current target set on the support set. Denoting the inner-level optimum obtained with the current weights by $\hat{\omega}^\star$, the outer-level gradient is given by:

$$\mathcal{I}(\theta_j^{\text{tar}},\theta^{\text{sup}};\hat{\omega}^*,\mathbf{v}) = -\left(\nabla_\omega\hat{R}(\theta^{\text{sup}};\hat{\omega}^*)\right)^\top H_{\hat{\omega}^*,\mathbf{v}}^{-1}\nabla_\omega\hat{R}(\theta_j^{\text{tar}};\hat{\omega}^*). \quad (13)$$

For brevity, we denote the empirical risk $\mathbb{E}[\ell(\cdot)]$ by $R$. $\nabla_\omega\hat{R}(\cdot;\hat{\omega}^*)$ denotes the gradient of the empirical risk with respect to the parameter $\omega$, while $H_{\hat{\omega}^*,\mathbf{v}}$ is the Hessian of the weighted objective evaluated at $\hat{\omega}^*$. The overall algorithm of BFR can be summarized in Algorithm 1.

## 3.4 COMPLEXITY ANALYSIS

Replacing the classifier with a Bayesian last layer in pre-training adds only a few parameters; the extra KL terms scale as $\mathcal{O}(C_{out}^{(L)})$ and $\mathcal{O}(C_{out}^{(L)}C)$, negligible next to a ResNet's $\mathcal{O}(\sum_l C_{in}^{(l)}C_{out}^{(l)}H^{(l)}W^{(l)}(K^{(l)})^2)$ cost. During retraining, BFR updates only the last layer. Let $m$ denote the total number of parameters in the Bayesian last layer, $m_{\text{full}}$ the number of parameters in the entire model, and $n$ the number of training samples, with $m_{\text{full}}\gg m$. Compared with other

last-layer retraining methods, BFR's additional cost primarily arises from the implicit differentiation used in the outer loop; since the number of outer iterations $K$ is typically small, the outer-loop complexity is $\mathcal{O}\big((n_{\text{support}} + n_{\text{target}})m + Km\big) \approx \mathcal{O}(nm)$. Overall, BFR has a obvious lower cost than full-model retraining methods (Liu et al., 2021a; Li et al., 2024) with complexity $\mathcal{O}(n\,m_{\text{full}})$; compared with standard last-layer retraining, it incurs only an extra cost linear in $m$. Experiment on computation complexity can be found in the Table 18 in Appendix D.6

### 3.5 Theoretical Analysis

In this section, we establish convergence guarantees for BFR. Following Lorraine et al., implicit differentiation in the outer loop is valid under: $i$) the inner objective attains a stationary point, i.e., its gradient with respect to $\omega$ vanishes at the optimum ($\nabla_\omega R = 0$); and $ii$) the inner objective is twice continuously differentiable with an invertible Hessian. Guided by these requirements, we make the following assumption:

**Assumption 1** *For inner loop objective Eq. 12 $L_{in}$. Let $\omega$ denotes the parameters optimized in the inner loop, and $\mathbf{v}$ are treated as constants. Assume there exists a point $\hat{\omega}^*$ such that:*

1. *$\mathcal{L}_{\text{in}}$ is twice continuously differentiable with respect to $\omega$;*

2. *First-order stationarity holds: $\nabla_\omega \mathcal{L}_{\text{in}}(\hat{\omega}^*; \mathbf{v}) = 0$;*

3. *The Hessian is invertible: $\nabla_\omega^2 \mathcal{L}_{\text{in}}(\hat{\omega}^*; \mathbf{v})$ is a nonsingular matrix.*

Under this assumption, the implicit function theorem can be invoked to characterize the differentiability of the inner optimum $\hat{\omega}^*(\mathbf{v})$ with respect to $\mathbf{v}$, and to derive an explicit expression for the outer-level gradient. Our inner objective is a sum of smooth terms (negative log-likelihood plus a KL regularizer) and is twice continuously differentiable with respect to $\omega$ under standard exponential-family assumptions and differentiable reparameterizations. By restricting the optimization to a bounded domain—an assumption also used in (Pedregosa, 2016; Lorraine et al., 2020)—the existence of an interior optimum is ensured, yielding the stationarity condition $\nabla_\omega \mathcal{L}_{\text{in}} = 0$. Under smoothness and invertibility conditions, the dependence of the outer gradient on the inner optimum can be formally expressed via the implicit function theorem, implying that the bi-level update indeed follows the descent direction of the worst-case risk under the induced uncertainty distribution. In addition, strong convexity in a neighborhood of the optimum (or the inclusion of a small damping term) makes the Hessian positive definite and thus invertible(Boyd & Vandenberghe, 2004). Hence, Assumption 1 is both reasonable and standard in practice. We further provide **an ablation of the bi-level optimization module in Figure 10**, assessing convergence via a sensitivity analysis.

## 4 Experiments

**Datasets** We benchmark on six datasets that expose diverse spurious cues, including **image datasets**: 1) Waterbirds (Wah et al., 2011) couples bird type with background, 2) CelebA (Liu et al., 2015) predicts blond hair amid a gender skew; **language datasets**: 1) MultiNLI (Williams et al., 2017) labels entailment/neutral/contradiction with contradiction tied to lexical negation, 2) CivilComments (WILDS) (Borkan et al., 2019) detects toxicity under severe identity imbalance; and **Distribution Shift**: ImageNet-9 (Xiao et al., 2021) and ImageNet-A (Hendrycks et al., 2021) provides naturally adversarial images to test ImageNet-9–trained models. Full setup is in Appx. B.

**Baselines** We compare to methods categorized based on their reliance on group annotation availability. *i)* **Group annotation for both Train and Val**: These methods rely on explicit group labels throughout training. We employ Group DRO (Sagawa et al., 2019) optimizes for worst-group performance by dynamically adjusting training weights based on group-wise loss. *ii)* **Group annotation for Val only:** These methods typically share the same ERM backbone, but additionally use group information on the validation set for a second-stage fine-tuning, including JTT (Liu et al., 2021b), DFR (Kirichenko et al., 2022), CnC (Zhang et al., 2022), AFR (Qiu et al., 2023), SELF (LaBonte et al., 2023), MAPLE (Zhou et al., 2022), GSR (Qiao et al., 2025) and DPE (To et al., 2025). *iii)* **Group annotation free:** These approaches do not require any explicit group labels and instead rely on instance-level signals or heuristics to mitigate spurious correlations, including BPA (Seo

Table 1: Average and worst-group test accuracies of different approaches evaluated on image datasets (Waterbird and CelebA) and natural language datasets (MultiNIL and CivilComments). We run **_BFR_** on 3 random seeds and report the mean and standard deviation. Best WGA are highlighted in bold, and the best result under the annotation-free setting is highlighted in bold$^\star$.

| Method | Group Info | | Waterbirds | | CelebA | | MultiNLI | | CivilComments | |
| --- | --- | --- | --- | --- | --- | --- | --- | --- | --- | --- |
| | Train | Val | Worst | Mean | Worst | Mean | Worst | Mean | Worst | Mean |
| ERM | - | - | $71.9_{\pm1.5}$ | $91.4_{\pm1.7}$ | $45.1_{\pm0.8}$ | $95.1_{\pm0.4}$ | $59.2_{\pm0.3}$ | $81.9_{\pm0.1}$ | $54.6_{\pm0.6}$ | $92.4_{\pm0.1}$ |
| ERM-BFR | - | - | $72.4_{\pm0.9}$ | $91.2_{\pm1.3}$ | $47.2_{\pm2.0}$ | $95.4_{\pm0.2}$ | $60.8_{\pm2.5}$ | $81.7_{\pm0.3}$ | $57.3_{\pm1.0}$ | $92.3_{\pm0.2}$ |
| Group DRO | Yes | Yes | 91.4 | 93.5 | 88.9 | 92.9 | 77.4 | 81.4 | 69.9 | 88.9 |
| JTT | No | Yes | 86.7 | 93.3 | 81.1 | 88.0 | 72.6 | 78.6 | 69.3 | 91.1 |
| DFR | No | Yes | $92.9_{\pm0.2}$ | $94.2_{\pm0.4}$ | $88.3_{\pm1.1}$ | $91.3_{\pm0.3}$ | $74.7_{\pm0.7}$ | $82.1_{\pm0.2}$ | $70.1_{\pm0.8}$ | $87.2_{\pm0.3}$ |
| CnC | No | Yes | $88.5_{\pm0.3}$ | $90.9_{\pm0.1}$ | $88.8_{\pm0.9}$ | $89.9_{\pm0.5}$ | - | - | $68.9_{\pm2.1}$ | $81.7_{\pm0.5}$ |
| AFR | No | Yes | $90.4_{\pm1.1}$ | $94.2_{\pm1.2}$ | $82.0_{\pm0.5}$ | $91.3_{\pm0.3}$ | $73.4_{\pm0.6}$ | $81.4_{\pm0.2}$ | $68.7_{\pm0.6}$ | $89.8_{\pm0.6}$ |
| SELF | No | Yes | $\mathbf{93.0}_{\pm0.3}$ | $94.0_{\pm1.7}$ | $83.9_{\pm0.9}$ | $91.7_{\pm0.4}$ | $70.7_{\pm2.5}$ | $81.2_{\pm0.7}$ | $79.1_{\pm2.1}$ | $87.7_{\pm0.6}$ |
| MAPLE | No | Yes | 91.7 | 92.9 | 88.0 | 89.0 | 72.7 | 77.2 | 64.1 | 89.7 |
| GSR | No | Yes | $92.9_{\pm0.0}$ | $94.0_{\pm1.7}$ | $87.0_{\pm0.4}$ | $90.0_{\pm0.0}$ | $\mathbf{78.5}_{\pm0.3}$ | $79.8_{\pm0.0}$ | $71.7_{\pm0.6}$ | $85.9_{\pm0.4}$ |
| DPE | No | Yes | $91.0_{\pm0.5}$ | $92.5_{\pm0.2}$ | $81.9_{\pm0.3}$ | $89.8_{\pm0.2}$ | $\underline{69.3}_{\pm0.8}$ | $81.3_{\pm0.2}$ | $69.9_{\pm0.9}$ | $82.2_{\pm0.2}$ |
| LfF | No | No | 78.0 | 91.2 | 77.2 | 85.1 | 70.2 | 80.8 | 58.8 | 58.8 |
| BPA | No | No | 71.4 | - | 82.5 | - | - | - | - | - |
| GEORGE | No | No | 76.2 | 95.7 | 52.4 | 94.8 | - | - | - | - |
| BAM | No | No | $89.1_{\pm0.2}$ | $91.4_{\pm0.3}$ | $80.1_{\pm3.3}$ | $88.4_{\pm2.3}$ | $70.8_{\pm1.5}$ | $80.3_{\pm1.0}$ | $79.3_{\pm2.7}$ | $88.3_{\pm0.8}$ |
| **BFR (Ours)** | No | No | $\mathbf{92.8}^{\star}_{\pm1.5}$ | $95.1_{\pm0.4}$ | $\mathbf{83.9}^{\star}_{\pm1.0}$ | $91.2_{\pm0.7}$ | $\mathbf{73.8}^{\star}_{\pm0.9}$ | $80.8_{\pm0.5}$ | $\mathbf{80.5}^{\star}_{\pm1.6}$ | $88.2_{\pm0.3}$ |

et al., 2022), GEORGE (Sohoni et al., 2020) and BAM (Li et al., 2023). Our method, BFR, also belongs to this category In the experiment on Imagent-9 and Imagnet-A, we introduce additional baselines including StylisedIN (Geirhos et al., 2018), RUBi (Cadene et al., 2019), ReBias (Bahng et al., 2020), LfF (Nam et al., 2020), CaaM (Wang et al., 2021), SSL+ERM (Kim et al., 2022) and LWBC (Kim et al., 2022). Per group-robustness research tradition, we report results from previous papers if available. More details for baselines can be found in Appendix B.1.

**Training Details** We first construct a modular ablation of BFR, denoted **ERM-BFR**, which corresponds to the model obtained after the probabilistic pretraining stage. It can be viewed as BFR without the uncertainty-guided bi-level retraining stage. For vision we use a ResNet-50 pretrained on ImageNet-1k as the backbone. For text we use the BERT-base-uncased model (Devlin et al., 2019) pretrained on the BookCorpus and English Wikipedia. **Worst-Group Accuracy (WGA)** is used as the main metric for evaluating group robustness. For all datasets, we follow the standard splits. Model selection uses worst-class accuracy on a support subset of the validation set.

## 4.1 EFFECTIVENESS OF UNCERTAINTY GUIDING

To show that Probabilistic Pretraining yields usable, well-calibrated structure, we t-SNE visualize Waterbirds features of ERM-BFR in Fig. 2: the left panel is colored by group labels, and the right panel by predictive uncertainty. According to the dataset specification, Groups 1 and 2 are minority subgroups, and high-uncertainty regions co-locate with subsets of these groups. To further quantify the correlation between, we evaluate the ROC-AUC by treating minority groups as the positive class and majority groups as the negative class, using the instance-level predictive uncertainty as the scoring function. the AUC is **0.87**, which is greater than 0.5 indicates performance above random chance and therefore reflects non-trivial information content (Hendrycks & Gimpel, 2017). We further report **calibration (ECE) in Table 11**. Taken together, this indicates Stage-1 uncertainty reliably flags under-represented groups and provides a sound guidance for the subsequent fine-tuning.

**Improvement in Group Robustness** (1) We benchmark BFR against the group-robustness baselines that use no group labels and also against semi-supervised methods that need labels only in validation. **Table 1** shows that, across vision and NLP datasets, BFR achieves the highest worst-group accuracy among group-annotation-free methods and remains competitive with semi-supervised ones (e.g., best WGA on CivilComments). This indicates that, by relying solely on model-derived predictive uncertainty, BFR can substantially improve worst-group performance without incurring any additional annotation cost. Compared to group-aware or semi-supervised methods that require group labels, BFR remains competitive and is even superior in some cases.

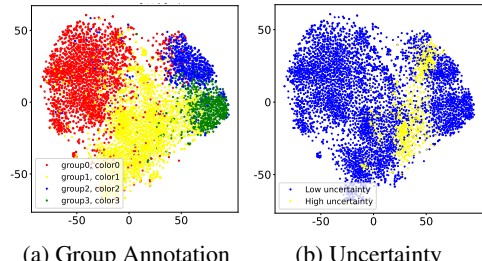

(a) Group Annotation  (b) Uncertainty

Figure 2: t-SNE vis of ERM-BFR feature emb shows that the predictive uncertainty **(b)** effectively highlights minority groups (with 0.87 AUC). **(a)** present in the data.

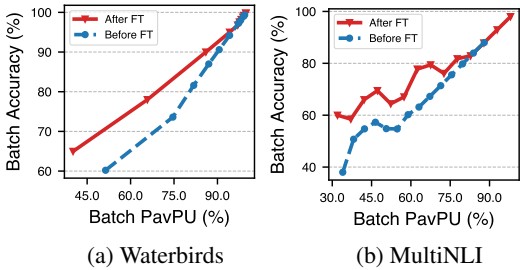

(a) Waterbirds  (b) MultiNLI

Figure 3: PavPu-Accuracy curves for **(a)** Waterbirds and **(b)** MultiNLI datasets. Red: after uncertainty-guided bi-level optimization; blue: before.

For example, on Waterbirds, BFR achieves a WGA of 92.8% without accessing any group labels, which is comparable to or slightly higher than Group DRO (91.4%) and DFR (92.9%), both of which use group information. Notably, BFR depends only on the predictive uncertainty provided by the Bayesian last layer, rather than explicit group labels, yet matches or exceeds meta-training–based methods (e.g., MAPLE, GSR) on most tasks and outperforms MAPLE on three of them. **Overall, under a fully group-annotation–free setting, BFR significantly outperforms existing annotation-free baselines and approaches or even surpasses state-of-the-art methods that rely on group labels**, providing strong evidence for the practical effectiveness of the proposed uncertainty-guided bi-level optimization framework in real-world label-scarce scenarios. (2) **Distribution Shift**: To probe BFR's resilience to distribution shift, we train it on ImageNet-9 and evaluate it on the much harder ImageNet-A benchmark. We use a ResNet-18 backbone initialized with standard ImageNet-1K weights and perform no additional fine-tuning on ImageNet-A. In this fully out-of-distribution setting, BFR attains 40.3% top-1 accuracy on ImageNet-A—the best among all compared methods—and 92.2% on ImageNet-9, as summarized in **Table 2**. Most importantly, it lowers the ImageNet-9 to ImageNet-A accuracy gap to 51.9%, a 14-point reduction relative to vanilla ERM (65.9%) and the smallest gap across the entire slate of baselines. These results show that the uncertainty-guided retraining in BFR not only preserves in-distribution performance but also markedly improves robustness to challenging out-of-distribution samples.

**Relation between Uncertainty and Accuracy**  We demonstrate that uncertainty guidance improves model performance in the second phase. Specifically, we use the PavPU metric(Mukhoti & Gal, 2018) (details in Appendix B) to compute batch-level uncertainties. The test set is sorted by the instance uncertainty ($p$-value) into subsets; for each subset we compute the average accuracy and PavPU and plot the results in Fig. 3. We observe: *i)* After uncertainty-guided fine-tuning, the model attains higher accuracy for the same level of uncertainty; *ii)* The improvements are concentrated in the high-uncertainty region: in subsets with lower PavPU, the gap between the red and blue curves is larger, indicating that uncertainty-guided fine-tuning helps hard samples more. Overall, after fine-tuning the curve shifts upward, showing that while maintaining certainty, accuracy is increased, verifying the effectiveness and robustness of uncertainty-guided fine-tuning. Additional ablation studies examining the contributions of different components and training stages are provided in Appendix D. As detailed in Appendix D, these ablations show that (i) full predictive uncertainty is consistently more effective than using only aleatoric or epistemic components, (ii) the method is robust to the choice of the $p$-value threshold, and (iii) both the probabilistic pretraining stage and the bi-level optimization contribute non-trivially to the final robustness gains.

## 4.2 EDITABILITY EVALUATION

Table 3 reports the fraction of parameters with $|\Phi_{2,i} - \Phi_{1,i}| > \tau$ (lower is better). For $\tau \in \{0, 10^{-4}, 10^{-3}, 10^{-2}\}$, we compute $\#\{i : |\Phi_{2,i} - \Phi_{1,i}| > \tau\}/|\Phi|$, a proxy for update selectivity (Wang et al., 2024). Using DFR (Kirichenko et al., 2022) as a conventional decision-layer baseline, BFR yields consistently smaller fractions on WaterBirds, CelebA, MultiNLI, and CivilComments for $\tau \leq 10^{-3}$, indicating its sparse head concentrates changes on task-relevant parameters while limiting collateral perturbations. At $\tau = 10^{-2}$, the gap on CivilComments narrows, likely because BFR's sparsity concentrates larger, targeted updates on a small set of key parameters.

Figure 4: GradCAM (Selvaraju et al., 2017) visulizations for BFR. **The second row** shows the visualizations after the pre-training stage, and **the third row** shows the results after uncertainty-guided fine-tuning. Spuriously correlated samples are identified by high uncertainty and are corrected after fine-tuning; samples with low uncertainty consistently focus on the core features and exhibit an even more concentrated pattern after fine-tuning.

Table 2: Average accuracy (%) and accuracy gap (%) on ImageNet-9 and ImageNet-A (ResNet-18 backbone). Best results are in **bold**.

| Method | ImageNet-9 | ImageNet-A | Acc. Gap ($\downarrow$) |
|---|---|---|---|
| ERM | $90.8_{\pm0.6}$ | $24.9_{\pm1.1}$ | 65.9 |
| ERM-BFR | $91.4_{\pm0.7}$ | $25.1_{\pm1.2}$ | 66.3 |
| StylisedIN | $88.4_{\pm0.5}$ | $24.6_{\pm1.4}$ | 63.8 |
| RUBi | $90.5_{\pm0.3}$ | $27.7_{\pm2.1}$ | 62.8 |
| ReBias | $91.9_{\pm1.7}$ | $29.6_{\pm1.6}$ | 62.3 |
| LfF | 86.0 | 24.6 | 61.4 |
| CaaM | **95.7** | 32.8 | 62.9 |
| SSL+ERM | $94.2_{\pm0.1}$ | $34.2_{\pm0.5}$ | 60.0 |
| LWBC | $94.0_{\pm0.2}$ | $36.0_{\pm0.5}$ | 58.0 |
| **BFR (Ours)** | $92.2_{\pm0.1}$ | $\mathbf{40.3_{\pm0.8}}$ | **51.9** |

Table 3: Parameter change (%) across thresholds. Smaller values indicate less parameter updates.

| Dataset | 0 | | 1e-4 | |
|---|---|---|---|---|
| | BFR | DFR | BFR | DFR |
| WaterBirds | **58.39** | 100.00 | **58.22** | 99.71 |
| CelebA | **52.88** | 100.00 | **52.83** | 99.83 |
| MultiNLI | **67.32** | 100.00 | **67.14** | 99.74 |
| CivilComments | **60.01** | 100.00 | **59.82** | 99.28 |

| Dataset | 1e-3 | | 1e-2 | |
|---|---|---|---|---|
| | BFR | DFR | BFR | DFR |
| WaterBirds | **56.39** | 96.58 | **28.70** | 67.18 |
| CelebA | **52.07** | 96.80 | **38.89** | 70.82 |
| MultiNLI | **65.54** | 96.96 | **49.63** | 67.98 |
| CivilComments | **58.91** | 91.29 | 48.76 | **34.98** |

**Visualization Result** We selected the top-5 samples with the highest and lowest uncertainty, respectively, and visualized their most highly activated features, as shown in Fig. 4 (See Appendix D.7 for more results). **The second row:** heatmaps for high-uncertainty samples often fall on background or spurious regions, whereas those for low-uncertainty samples primarily focus on key parts of the target. **The third row:** the attention of high-uncertainty samples is corrected, with heatmaps converging on semantically relevant objects; low-uncertainty samples further contract to a more concentrated core area. Overall, the results indicate that BFR **preserves originally correct activations while correcting neurons dominated by spurious correlations**, demonstrating its capability for uncertainty-guided, localized parameter updates.

## 5 CONCLUSION

We propose BFR, which integrates calibrated instance-level uncertainty into the training loop as a control signal and enforces sparse, localized parameter updates. BFR exploits both aleatoric and epistemic uncertainty of each input to guide a stable last-layer retraining procedure. Empirically, BFR demonstrates effectiveness in improving both group robustness and performance under distribution shift, and it is broadly applicable to both ResNet- and Transformer-based models. We substantiate that BFR performs sparse, localized parameter updates through quantitative analyses and visualizations, and we analyze its optimization stability both theoretically and empirically. Additional ablations isolate the contributions of each uncertainty component and of the retraining strategy. Taken together, these results suggest that BFR is an efficient component for robust learning pipelines.

ETHICS STATEMENT

Our study uses only publicly available research datasets under their original licenses; no new human data were collected and no personally identifiable information was accessed.

REPRODUCIBILITY STATEMENT

The novel methods introduced in this paper are accompanied by detailed descriptions (Sec. 3), and their implementations are provided at Supplementary materials.

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

# APPENDIX

## A    THE USE OF LARGE LANGUAGE MODELS (LLMS)

We used LLMs only for **light proofreading and formatting**; they were **not involved** in data annotation, modeling, or algorithmic decision-making.

---

**Algorithm 1** Bayesian Feature Reweighting

---

1: **Input**: Training dataset $D_{train}$, validation dataset $D_{val}$, outer loop steps $K$, number of epochs $L$, model parameters $\omega = (\theta, \Phi, \omega_{rest})$.
2: **Output**: Updated model parameters $\omega^{\star}$
3: *Stage 1: Probabilistic Pretraining*:
4: **for** epoch = 1 to $L$ **do**
5:     **for** each mini-batch $\mathcal{B}_k = \{(x_j, y_j)\}_{j=1}^{B}$ **do**
6:         Sample $\theta_j$ and $\Phi$ from posterior distribution $q(\theta_j|x_j)$ and $q(\Phi|-)$, respectively.
7:         Sample $y_j$ according to Eq. 5
8:         compute gradients $\nabla_{\omega}L$ according to Eq. 6
9:         Update: $\omega \leftarrow \nabla_{\omega}L$
10:     **end for**
11: **end for**
12: *Stage 2: Uncertainty-guided Bi-level Optimization*:
13: Calculate Uncertainty $U(.)$ for $D_{val}$ according to Eq. 9
14: Dividing $D_{val}$ to $D_{sup}$ and $D_{tar}$ according to $U(.)$
15: Initialize Support sample weights **v**
16: **for** iter = 1 to $K$ **do**
17:     **Inner-loop:**
18:     Sample $\theta_j$ from the optimal posterior $q^*(\theta_j^*|x_j)$
19:     Sample $\Phi$ from $q(\Phi|-)$
20:     Update $\omega$ according to Eq. 12
21:     **Outer-loop:** Update **v** according to Eq. 13
22: **end for**
23: **RETURN:** Updated model parameters $\omega^*$

---

## B    EXPRIMENTAL SETUP

All experiments are conducted on Linux servers equipped with 32 Intel(R) Xeon(R) Gold 6326 CPU @ 2.90GHz and 4 NVIDIA 4090 GPUs. Models are implemented in PyTorch version 2.0.1 and Python 3.9.

**Uncertainty evaluation metric.**    We estimate uncertainty using a hypothesis testing approach (Fan et al., 2021). To evaluate uncertainty estimates, we use the Patch Accuracy vs. Patch Uncertainty (PAvPU) metric (Mukhoti & Gal, 2018), which defined as PAvPU = $(n_{ac} + n_{iu})/(n_{ac} + n_{au} + n_{ic} + n_{iu})$, where $n_{ac}$, $n_{au}$, $n_{ic}$, and $n_{iu}$ represent the counts of accurate-certain, accurate-uncertain, inaccurate-certain, and inaccurate-uncertain samples, respectively. Higher PAvPU values indicate that the model reliably produces accurate predictions with high certainty and inaccurate ones with high uncertainty. The threshold for $p$-value is set to 0.05 in the experiments.

**Evaluation Metrics**    Robustness to spurious bias is measured with **worst-group accuracy (WGA)**, the minimum test accuracy achieved over all latent groups and therefore the model's lower-bound performance under dataset bias.

### B.1    DETAILS OF DATASETS AND BASELINES

Details of the dataset distributions are shown in Table 4.

Table 4: Data quantities and group compositions for each dataset (train / validation / test).

| Dataset | Class $y$ | Spurious $a$ | Train | Validation | Test |
|---------|-----------|--------------|-------|------------|------|
| *Waterbirds* | | | | | |
| Waterbirds | landbird | land | 3498 | 467 | 2225 |
| Waterbirds | landbird | water | 184 | 466 | 2225 |
| Waterbirds | waterbird | land | 56 | 133 | 642 |
| Waterbirds | waterbird | water | 1057 | 133 | 642 |
| *CelebA* | | | | | |
| CelebA | non-blond | female | 71629 | 8535 | 9767 |
| CelebA | non-blond | male | 66874 | 8276 | 7535 |
| CelebA | blond | female | 22880 | 2874 | 2480 |
| CelebA | blond | male | 1387 | 182 | 180 |
| *MultiNLI* | | | | | |
| MultiNLI | contradiction | no negation | 57498 | 22814 | 34597 |
| MultiNLI | contradiction | negation | 11158 | 4634 | 6655 |
| MultiNLI | entailment | no negation | 67376 | 26949 | 40496 |
| MultiNLI | entailment | negation | 1521 | 613 | 886 |
| MultiNLI | neither | no negation | 66630 | 26655 | 39930 |
| MultiNLI | neither | negation | 1992 | 797 | 1148 |
| *CivilComments* | | | | | |
| CivilComments | neutral | no identity | 148186 | 25159 | 74780 |
| CivilComments | neutral | identity | 90337 | 14966 | 43778 |
| CivilComments | toxic | no identity | 12731 | 2111 | 6455 |
| CivilComments | toxic | identity | 17784 | 2944 | 8769 |

- **Waterbirds** (Wah et al., 2011) is a binary image classification dataset that distinguishes between waterbirds and landbirds. A prominent spurious correlation exists between the bird type and the background environment, as waterbirds are predominantly associated with water scenes, while landbirds appear against land backgrounds.

- **CelebA** (Liu et al., 2015) is a binary facial attribute classification dataset focused on predicting hair blondness. In this case, a spurious correlation arises due to an imbalance in the co-occurrence of gender and hair color, with male individuals being less likely to be labeled as blond.

- **MultiNLI**(Williams et al., 2017) is a multiclass natural language inference dataset in which sentence pairs are labeled entailment, neutral, or contradiction. The contradiction class is spuriously associated with the presence of lexical negation, which can lead to overfitting to surface-level patterns.

- **CivilComments (WILDS)** (Borkan et al., 2019; Koh et al., 2021) is a binary text classification dataset aimed at detecting toxic comments. It includes eight identity attributes (e.g., gender, religion), and samples are grouped by identity and label, forming 16 overlapping subgroups. Although the dataset does not exhibit clearly defined spurious correlations, it suffers from significant group imbalance, particularly among non-toxic comments referencing marginalized identities such as minority religions.

- **ImageNet-9** (Xiao et al., 2021) is a subset of ImageNet (Deng et al., 2009) containing nine super-classes. It comprises images with different background and foreground signals and can be used to assess how much models rely on image backgrounds.

- **ImageNet-A**(Hendrycks et al., 2021) is a dataset of real-world images, adversarially curated to test the limits of classifiers such as ResNet-50. We used this dataset to test the robustness of a classifier after training it on ImageNet-9.

The baselines are described as follows:

- **Group annotation for both training and validation**: Group DRO (Sagawa et al., 2019) optimizes for worst-group performance by dynamically adjusting training weights based on group-wise loss.

- **Group annotation for validation only:** JTT (Liu et al., 2021b) identifies potentially underrepresented samples by upweighting high-loss training examples. DFR (Kirichenko et al., 2022) performs last-layer retraining using a group-balanced subset selected from the validation set. CnC (Zhang et al., 2022) further incorporates contrastive learning to align intra-class representations. AFR (Qiu et al., 2023) improves group robustness efficiently by automatically reweighting feature channels to enhance generalization to unseen or underrepresented groups. SELF (LaBonte et al., 2023) proposes a last-layer retraining approach that achieves group robustness with minimal group label annotations by strategically selecting training examples to improve performance on minority groups. MAPLE (Zhou et al., 2022) jointly reweights the training samples and finetunes the model using information from the validation set. GSR (Qiao et al., 2025) leverages influence functions to dynamically reweight training samples based on their contribution to group-generalizable performance, even with limited group supervision. DPE (To et al., 2025) derives pseudo group labels by leveraging a model's own clustering capacity to form fine-grained subgroups.

- **Group annotation free:** BPA (Seo et al., 2022) identifies spurious correlations through unsupervised feature attribution and mitigates them via targeted sample reweighting. GEORGE (Sohoni et al., 2020) uses learned group structure from pretrained representations to construct pseudo-groups for balanced training. BAM (Li et al., 2023) introduces a two-stage framework that detects biased predictions and retrains on bias-mitigated sample subsets without requiring group labels.

For the baselines used in Imagent-9 and Imagent-A, the details are as follows:

- **StylisedIN** (Geirhos et al., 2018): Data-level debiasing by training on a "stylized" version of ImageNet where texture cues are randomized via style transfer, encouraging models to rely on shape rather than spurious textures.

- **RUBi** (Cadene et al., 2019): Adds a bias-only branch whose output gates the main classifier's logits; samples that the bias branch can solve get down-weighted, reducing reliance on shortcut cues.

- **ReBias** (Bahng et al., 2020): Trains a bias-capturing network alongside the main model and enforces feature separation (via adversarial/orthogonality losses), so the main model learns bias-agnostic representations.

- **LfF** (Learning from Failure) (Nam et al., 2020): First fits a biased model, then up-weights (or re-samples) the examples it gets wrong—"bias-conflicting" samples—to guide the final model toward robust features.

- **CaaM** (Wang et al., 2021): Introduces a causal attention/activation module that intervenes on feature maps to suppress spurious regions and emphasizes causally relevant evidence, mitigating bias at the feature level.

- **SSL+ERM**(Kim et al., 2022): Uses self-supervised pretraining to acquire more general features, then applies standard ERM fine-tuning, which empirically reduces dependence on dataset biases.

- **LWBC** (Kim et al., 2022): A two-stage procedure that estimates bias-confounding patterns and rebalances training (e.g., via sample reweighting or batch composition), enabling "learning without biased cues."

### B.2 HYPERPARAMETERS

The hyperparameters we used are listed in Table 5, 7 and 6.

Table 5: Hyperparameters for base model training.

| Dataset | optimizer | lr | scheduler | batch size | weight decay | epochs |
|---|---|---|---|---|---|---|
| Waterbirds | SGD | 1e-3 | Cosine | 32 | 1e-3 | 100 |
| CelebA | SGD | 1e-3 | Cosine | 128 | 1e-4 | 50 |
| MultiNLI | AdamW | 1e-5 | Cosine | 16 | 1e-4 | 5 |
| CivilComments | AdamW | 1e-5 | Cosine | 16 | 1e-4 | 5 |
| Imagenet-9 | SGD | 1e-4 | Cosine | 128 | 1e-4 | 120 |
| Imagenet-A | SGD | 1e-4 | Cosine | 128 | 1e-4 | 120 |

Table 6: Hyperparameters for sample weight updates (outer loop).

| Dataset | optimizer | lr | scheduler | grad clip | $\tau$ | steps |
|---|---|---|---|---|---|---|
| Waterbirds | GD | 1 | StepLR | 1 | 0.1 | 100 |
| CelebA | GD | 1 | StepLR | 1e-2 | 0.01 | 100 |
| MultiNLI | GD | 1 | StepLR | 1e-2 | 1 | 50 |
| CivilComments | GD | 1 | StepLR | 1e-2 | 0.1 | 50 |
| Imagenet-9 | GD | 1 | StepLR | 1 | 0.1 | 100 |
| Imagenet-A | GD | 1 | StepLR | 1 | 0.1 | 100 |

## C  ELBO INSTANTIATION AND THE TRAINING-TIME UNCERTAINTY SIGNAL

**Scope.** This appendix *instantiates* the standard Evidence Lower Bound (ELBO) to our setting in order to (i) define the uncertainty signal used as a *training-time control variable* in Sec. 3.3, and (ii) state the regularity conditions that support the outer-gradient differentiability in Sec. 3.5. Algebraic details that are common to variational inference are omitted for brevity and can be found in (Kingma et al., 2015; Rezende et al., 2014; Hoffman et al., 2013; Bishop & Nasrabadi, 2006).

### C.1  MODEL AND STANDARD ELBO.

Let $x \in \mathcal{X}$, $y \in \mathcal{Y}$. We use a probabilistic head with *local* latent variables $\theta$ (instance-specific; modeling aleatoric uncertainty) and *global* latent variables $\Phi$ (shared; modeling epistemic uncertainty). Given parameters $\omega$, the variational family factorizes as $q_\omega(\theta, \Phi \mid x) = q_\omega(\theta \mid x, \Phi)\, q_\omega(\Phi)$. The standard ELBO for $\log p_\omega(y \mid x)$ writes

$$\mathcal{L}_{\text{ELBO}}(x, y; \omega) = \mathbb{E}_{q_\omega(\theta, \Phi \mid x)}\big[ \log p_\omega(y \mid x, \theta, \Phi) \big] - \text{KL}\big(q_\omega(\theta, \Phi \mid x) \,\|\, p(\theta, \Phi)\big), \quad (14)$$

which we adopt to obtain calibrated posteriors without claiming novelty in the ELBO form itself. Eq. equation 14 corresponds to Eq. (6) in the main text and follows the canonical reconstruction+KL structure.

### C.2  FROM ELBO TO AN UNCERTAINTY CONTROL SIGNAL.

The posterior predictive is

$$p_\omega(y \mid x) = \int p_\omega\big(y \mid x, \theta, \Phi\big)\, q_\omega(\theta, \Phi \mid x)\, d\theta\, d\Phi. \quad (15)$$

Table 7: Hyperparameters for last-layer retraining (inner loop).

| Dataset | optimizer | lr | scheduler | batch size | weight decay | epochs | uncertainty threshold |
|---|---|---|---|---|---|---|---|
| Waterbirds | Adam | 5e-4 | Constant | 64 | 1e-1 | 100 | 0.05 |
| CelebA | Adam | 5e-4 | Constant | 128 | 1e-2 | 100 | 0.05 |
| MultiNLI | Adam | 5e-4 | Constant | full | 1e-2 | 100 | 0.05 |
| CivilComments | Adam | 5e-4 | Constant | full | 1e-3 | 150 | 0.05 |
| Imagenet-9 | Adam | 1e-4 | Constant | full | 1e-1 | 150 | 0.05 |
| Imagenet-A | Adam | 1e-4 | Constant | full | 1e-1 | 150 | 0.05 |

We define an instance-level uncertainty score $U(x)$ from $p_\omega(y \mid x)$ (as described in Sec. 3.3). Given a threshold $\delta$, we induce the *uncertainty-conditioned* partition

$$\mathcal{D} = \mathcal{D}_{\text{sup}} \cup \mathcal{D}_{\text{tar}}, \qquad \mathcal{D}_{\text{sup}} = \{x : U(x) < \delta\}, \quad \mathcal{D}_{\text{tar}} = \{x : U(x) \geq \delta\}, \qquad (16)$$

which supplies the **support/target** sets used as the control signal in the bi-level objective of Sec. 3.3 (Eqs. 10–11). Hence, Eq. equation 14 is *functionally instantiated* to produce $U(x)$ for training-time control.

## C.3 What we omit (and why).

We intentionally *omit* generic ELBO algebra (Jensen steps, integral-to-expectation rewrites, and KL rearrangements), as they are not specific to our method and can be found in standard references (Kingma et al., 2015; Rezende et al., 2014; Hoffman et al., 2013; Bishop & Nasrabadi, 2006). What is *retained* are exactly the pieces needed to (i) map Eq. equation 14 to the uncertainty signal $U(x)$ (Eq. equation 16), and (ii) state the regularity conditions that connect Sec. 3.2 to the bi-level control objective (Sec. 3.3) and the outer-gradient analysis (Sec. 3.5).

## C.4 Discussion with previos works

Regarding Fan et al. (2021), the $p$-value in Contextual Dropout is used to measure differences between dropout activations, serving the purpose of enhancing sample-dependent dropout behavior. In our work, the $p$-value is used solely as a statistical testing tool to assess whether the top-2 posterior predictive probabilities differ significantly, which in turn provides a principled control signal for forming the support and target sets. Moreover, unlike Fan et al. (2021), our uncertainty arises from posterior sampling in a probabilistic last layer, with a clear aleatoric and epistemic decomposition and well-defined probabilistic semantics. This gives the resulting partition and the bi-level updates a more explicit statistical grounding. **Our framework does not incorporate contextual dropout, nor do we use $p$-values as part of network structure or regularization; therefore, the purpose and role of $p$-value in our method are fundamentally different from those in Fan et al. (2021).**

Regarding Zhou et al. (2022), our inner–outer objectives are not based on MAPLE-style approximate backpropagation-through-trajectory. Instead, we employ differentiable implicit gradients, allowing the influence of sample weights to propagate only through the inner-level optimum, which ensures numerical stability in the absence of group supervision. Furthermore, unlike Zhou et al. (2022), our inner loop optimizes only the Bayesian Last Layer and does so under sparsity, enabling localized and editable updates that substantially limit parameter perturbation (as shown in Table 3). We additionally report a direct comparison of computational cost in Table18, showing that our approach is significantly more efficient than Zhou et al. (2022).

### C.4.1 Discussion about Spurious Correlation

In addressing spurious correlations and subpopulation shifts, a large body of work has approached the problem from the perspective of group robustness, leveraging explicit group annotations to improve worst-group performance. Typical methods treat groups—defined as combinations of class labels and spurious attributes—as the basic units, and either directly minimize the worst-group loss or apply group-based reweighting and regularization during training Sagawa et al. (2020); Michel et al. (2022), or use group labels to reweight features and retrain the last layer to reduce reliance on non-causal cues Idrissi et al. (2022); Kirichenko et al. (2022). However, obtaining such fine-grained group annotations in practice often requires additional domain knowledge or metadata, making them expensive and difficult to scale. As illustrated in Fig. 5, group annotations specify not only the semantic class (e.g., "Waterbird" vs. "Landbird") but also the environmental or attribute context in which the sample appears (e.g., "Waterbird on water" vs. "Waterbird on land"). Such fine-grained subpopulation labels require identifying class–background combinations and are therefore substantially more costly to obtain than standard class labels. To alleviate this, recent work has explored maintaining subpopulation robustness while substantially reducing, or partially removing, the need for group labels Qiu et al. (2023); Liu et al. (2021a); Qiao et al. (2025); To et al. (2025); Li et al. (2024). Nonetheless, most of these approaches still rely on a non-trivial amount of group supervision for model selection, or on explicitly identifying minority subgroups and repeatedly retraining the full network, which introduces considerable group annotation and computational overhead and

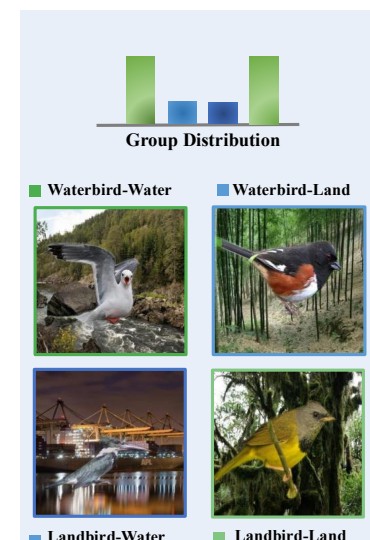

Figure 5: An illustration of group annotations in the Waterbirds dataset. Each image belongs to a group defined by the combination of its class (Waterbird vs. Landbird) and background environment (Water vs. Land). These fine-grained labels identify subpopulations that differ in spurious correlations, but are costly to obtain in practice.

may still struggle to generalize to unseen or unlabeled subpopulations Yang et al. (2023); Zhang et al. (2022).

In contrast to these methods, we start from the perspective of uncertainty estimation. On the one hand, uncertainty is inherently independent of explicit group labels and has been widely used to characterize near-boundary or out-of-distribution samples Hüllermeier & Waegeman (2021), providing a transferable signal for identifying high-risk subgroups in group-annotation-free settings. On the other hand, uncertainty estimates obtained under standard ERM are notoriously over-confident Guo et al. (2017), and directly weighting the loss by uncertainty often leads to optimization instability Kendall & Gal (2017); Seitzer et al. (2022). To address this, we replace the conventional softmax head with a Bayesian Last Layer Weber et al. (2018); Hu et al. (2025) to obtain calibrated uncertainty with sparsity-inducing priors, and we couple this layer with a bi-level optimization framework Ren et al. (2018); Qiao et al. (2025) for sample reweighting. This design allows us to completely remove dependence on group labels throughout both training and model selection, while confining the optimization cost to localized updates of the last layer. Under fully group-annotation-free information constraints, our approach thus achieves substantial improvements in worst-group performance at relatively low computational cost, which constitutes a key distinction and advantage over existing methods for improving group robustness.

### C.4.2    DISCUSSION ABOUT THE EFFECTIVENESS OF LAST LAYER FINE-TUNING

In our framework, the second stage is deliberately designed as a localized adaptation step on top of a pretrained backbone, rather than full model retraining. This is in line with a common paradigm in recent work on group robustness and distribution shift, where a pretrained backbone is frozen and only the last layer is retrained or reweighted (e.g., DFR (Kirichenko et al., 2022), GSR(Qiao et al., 2025)), achieving strong worst-group performance with low computational overhead.

Empirically, this localized adaptation does not appear to impose a performance bottleneck. As shown in Table 1 of the main text, BFR, which only updates the Bayesian last layer, already outperforms several methods that update many more parameters or approximate full-parameter fine-tuning (e.g. JTT (Liu et al., 2021a), CNC (Zhang et al., 2022), BAM (Li et al., 2023)) across all four benchmarks in terms of worst-group accuracy.

To further probe the trade-off between adaptation scope and robustness within our own framework, we conduct an ablation in which we gradually unfreeze more layers in Stage 2 while keeping the

uncertainty-guided bi-level optimization scheme unchanged. On Waterbirds, we denote by "0" the default BFR setting (only the last layer is updated), and by "1–4" the variants that unfreeze the last 1–4 additional layers of the backbone. The results are summarized in Table 8.

Table 8: Effect of increasing the number of trainable layers in Fine-tuning stage on Waterbirds. "0" corresponds to the default BFR setting where only the Bayesian last layer is updated, while "1–4" denote variants that additionally unfreeze the last 1–4 backbone layers. We report worst-group accuracy (WGA) and mean accuracy (ACC).

| Metric | Number of unfrozen layers in Stage 2 | | | | |
|---|---|---|---|---|---|
| | 0 (last layer) | 1 | 2 | 3 | 4 |
| WGA | 92.43 | 86.60 | 87.23 | 85.83 | 85.83 |
| ACC | 94.63 | 95.51 | 95.10 | 95.18 | 95.18 |

Unfreezing additional layers leaves the mean accuracy essentially unchanged (or slightly higher), but consistently reduces worst-group accuracy by about 5–7 percentage points compared to the last-layer-only BFR. This suggests that, under our uncertainty-guided bi-level optimization and backbone configuration, keeping the backbone fixed acts as a useful regularizer that preserves its representation while allowing the Bayesian last layer to perform targeted corrections.

From the computational perspective, the complexity analysis in Sec. 3.4 and Appendix D.6 further illustrates the efficiency aspect of this design. On Waterbirds, a full-parameter fine-tuning method such as BAM requires substantially longer total training time (57.32 minutes) than BFR (16.33 minutes) under the same hardware and training setup, while still underperforming BFR in worst-group accuracy on all four datasets. Overall, these results indicate that restricting Stage 2 updates to the Bayesian last layer offers a favorable robustness–efficiency trade-off within our framework, providing strong robustness improvements with localized parameter changes and moderate computational cost.

# D  ADDITIONAL EXPERIMENT RESULTS

To better understand the sources of robustness gains in BFR, we conduct a set of ablation studies that isolate the contributions of different uncertainty components and training stages: (1) we decompose the predictive uncertainty used in our bi-level update into its aleatoric and epistemic parts (**See appendix D.1**); (2) We analyze the effect of the $p$-value threshold used for significance-based sample partitioning (**See sppendix D.3**); (3) We we evaluate the effectiveness of the probabilistic pretraining stage by comparing ERM-BFR with standard ERM (**Table 11**), followed by the ablations of uncertainty proxy (**See Table 12, 13**), sparsity prior (**See Table 15, 14**), and an assessment of the bi-level optimization stage through comparisons with direct uncertainty weighting (**Figure 10**); (4) Finally, we provide a complexity analysis to quantify the computational overhead of BFR relative to existing methods (**See Table 18**). Together, these ablations offer a detailed characterization of how each component contributes to the overall performance and stability of BFR.

## D.1  ABLATION: EFFECT OF PREDICTIVE VS ALEATORIC/EPISTEMIC UNCERTAINTY

To evaluate the effects of the aleatoric uncertainty and epistemic uncertainty separately, we have conducted additional ablation studies to examine their independent contributions. Specifically, we separately employ the aleatoric uncertainty (captured by the local latent variable $\theta$) and the epistemic uncertainty (captured by the global latent variable $\Phi$) as the control signal in the bi-level optimization, while keeping all other settings unchanged as Table 1. Using the same datasets and evaluation protocol as in the main text, we compare the Worst-Group Accuracy (WGA) and Mean Accuracy (Mean) under each variant. The results are summarized in the following table 9 and 10.

From the table, we observe that across multiple benchmarks, using either aleatoric or epistemic uncertainty alone already yields non-trivial robustness gains, and their effects are complementary: combining them into the full predictive uncertainty consistently achieves the best worst-group accuracy, with an average improvement of about 5–7 percentage points. This indicates that, in our framework, the use of predictive uncertainty is not a simple additive combination of aleatoric and

epistemic terms, but rather a natural exploitation of their joint posterior, which is more effective for identifying and emphasizing high-risk samples during training. In addition, we distinguish aleatoric and epistemic uncertainty in the modelling part because our posterior modeling is explicitly built on these two types of uncertainty, while the predictive uncertainty used during training is precisely their unified form.

Table 9: Ablation on different uncertainty types for image datasets (Waterbirds and CelebA). Results are reported as mean $\pm$ std over 3 runs.

| Uncertainty | Waterbirds | | CelebA | |
|---|---|---|---|---|
| | WGA | Mean | WGA | Mean |
| ERM | $71.9 \pm 1.5$ | $91.4 \pm 1.7$ | $45.1 \pm 0.8$ | $95.1 \pm 0.4$ |
| Aleatoric ($\theta$) | $85.98 \pm 0.6$ | $95.58 \pm 0.3$ | $78.33 \pm 0.4$ | $92.26 \pm 0.2$ |
| Epistemic ($\Phi$) | $87.38 \pm 0.9$ | $95.65 \pm 0.3$ | $68.89 \pm 0.6$ | $92.93 \pm 0.3$ |
| Predictive (BFR) | $\mathbf{92.8 \pm 1.5}$ | $95.1 \pm 0.4$ | $\mathbf{83.9 \pm 1.0}$ | $91.2 \pm 0.7$ |

Table 10: Ablation on different uncertainty types for language/text datasets (MultiNLI and Civil-Comments). Results are reported as mean $\pm$ std over 3 runs.

| Uncertainty | MultiNLI | | CivilComments | |
|---|---|---|---|---|
| | WGA | Mean | WGA | Mean |
| ERM | $59.2 \pm 0.3$ | $81.9 \pm 0.1$ | $54.6 \pm 0.6$ | $91.4 \pm 0.1$ |
| Aleatoric ($\theta$) | $70.12 \pm 0.8$ | $80.60 \pm 0.2$ | $67.91 \pm 0.7$ | $91.41 \pm 0.3$ |
| Epistemic ($\Phi$) | $70.91 \pm 0.9$ | $80.55 \pm 0.3$ | $73.64 \pm 0.8$ | $90.32 \pm 0.2$ |
| Predictive (BFR) | $\mathbf{73.8 \pm 0.9}$ | $80.8 \pm 0.5$ | $\mathbf{80.5 \pm 1.6}$ | $88.2 \pm 0.3$ |

## D.2 ABLATION: EFFECT OF PROBABILISTIC PRETRAINING

To demonstrate that ERM-BFR provides better calibration at the probabilistic pretraining stage, we report ECE (Naeini et al., 2015) on Waterbirds, CelebA, MultiNLI, and CivilComments (Table 11). Each model is trained with three random seeds, and we report the mean and standard deviation. ERM-BFR consistently achieves lower ECE than ERM across all four datasets while maintaining comparable accuracy, and even outperforms ERM in accuracy on some datasets.

Table 11: We evaluate the uncertainty estimation quality of ERM-BFR and compare it against ERM, reporting Expected Calibration Error (ECE), where lower values indicate better calibration.

| Model | Waterbirds | | CelebA | |
|---|---|---|---|---|
| | ACC | ECE | ACC | ECE |
| ERM | $91.4 \pm 1.7$ | $0.0607 \pm 0.01$ | $95.1 \pm 0.4$ | $0.0379 \pm 0.02$ |
| ERM-BFR | $\mathbf{92.0 \pm 1.7}$ | $\mathbf{0.0552 \pm 0.01}$ | $\mathbf{95.4 \pm 0.2}$ | $\mathbf{0.0190 \pm 0.01}$ |

| Model | CivilComments | | MultiNLI | |
|---|---|---|---|---|
| | ACC | ECE | ACC | ECE |
| ERM | $91.4 \pm 0.1$ | $0.0556 \pm 0.01$ | $\mathbf{81.9 \pm 0.1}$ | $0.1156 \pm 0.02$ |
| ERM-BFR | $\mathbf{92.1 \pm 0.2}$ | $\mathbf{0.0542 \pm 0.01}$ | $81.7 \pm 0.3$ | $\mathbf{0.1091 \pm 0.02}$ |

**Effect of different Uncertainty Proxy.** To demonstrate the importance of the probabilistic pre-training component, it is necessary to compare against alternative uncertainty estimation methods. Here, we conducted the following experiments: (1) Comparison of calibration quality and robustness gains. We include three representative baselines: (i) a loss-based hard-sample–selection method (Nam et al., 2020)—also included in Table 1 of the main paper—which identifies bias-conflicting

samples using a bias-only model and therefore represents a canonical loss-driven uncertainty proxy; (ii) a confidence-based proxy using the entropy of ERM outputs; and (iii) an dropout-based uncertainty proxy using dropout sampling. To ensure fairness, all proxies use the same ERM backbone, and we only replace the uncertainty estimation method. We then evaluate worst-group accuracy (WGA) and mean accuracy on the Waterbirds and CelebA datasets, following the same experimental protocol as Table 1. The results are in Table 12:

Table 12: Comparison of different uncertainty proxies used to guide the second-stage optimization. We report worst-group accuracy (WGA) and mean accuracy on Waterbirds and CelebA. All methods share the same ERM backbone, and results are reported as mean $\pm$ standard deviation over three random seeds.

| Uncertainty Proxy | Waterbirds | | CelebA | |
|---|---|---|---|---|
| | WGA | Mean ACC | WGA | Mean ACC |
| LfF (loss-based) | 78.0 | 91.2 | 77.2 | 85.1 |
| Entropy-based | $88.2 \pm 1.1$ | $96.2 \pm 0.5$ | $76.1 \pm 0.9$ | $92.0 \pm 0.4$ |
| Dropout-based | $87.2 \pm 0.8$ | $95.9 \pm 0.5$ | $68.3 \pm 1.2$ | $93.0 \pm 0.6$ |
| **BFR (Ours)** | $\mathbf{92.8 \pm 1.5}$ | $\mathbf{95.1 \pm 0.4}$ | $\mathbf{83.9 \pm 1.0}$ | $\mathbf{91.2 \pm 0.7}$ |

As shown in the Table 12, uncertainty proxies based on loss, entropy, or dropout consistently achieve weaker worst-group performance than BFR, while their mean accuracy remains at a similar level. This indicates that although these alternative proxies can preserve overall accuracy, they are less effective at identifying bias-conflicting samples and improving worst-group accuracy. In contrast, BFR with a Bayesian last layer yields more stable and consistently stronger robustness gains across both datasets, suggesting that the uncertainty signal provided by BLL is more effective for subgroup partitioning and sample selection in our framework.

(2) Comparison of parameter editability. We further analyze the extent of parameter changes before and after uncertainty-guided bi-level optimization when using confidence-based and MC-based uncertainty proxies. Following the metric used in Table 3 of the main paper, we sample one run per model from multiple random runs of each Stage-1 proxy model, and compute the proportion of last-layer weights whose absolute change exceeds a given threshold on waterbirds datasets(lower values indicate more localized parameter updates). The results are shown in Table 13. The results show that entropy- and dropout-based proxies lead to substantially larger and more widespread parameter updates. In contrast, BFR due to its sparsity-inducing priors and structured posterior which modifies only a small, well-targeted subset of parameters during retraining, thus markedly improving editability and reducing collateral interference.

Table 13: Proportion (%) of last-layer weights on Waterbirds whose absolute change exceeds a threshold $\delta$ after the second-stage fine-tuning. Columns correspond to different thresholds $\delta \in \{0, 10^{-4}, 10^{-3}, 10^{-2}\}$. Lower values indicate more localized parameter updates.

| Uncertainty Proxy | Waterbirds | | | |
|---|---|---|---|---|
| | 0 | $10^{-4}$ | $10^{-3}$ | $10^{-2}$ |
| Entropy-based | 100 | 99.82 | 96.24 | 64.54 |
| Dropout-based | 100 | 99.85 | 97.68 | 75.81 |
| **BFR (Ours)** | **58.39** | **58.22** | **56.39** | **28.70** |

**Effect of the Sparsity Prior.** To examine the effect of the sparsity-inducing prior in the Bayesian last layer, we compare the Gamma prior used in the main paper (non-negative and sparsity-inducing) with a standard Gaussian prior $\mathcal{N}(0, 1)$ on the Waterbirds dataset. The backbone, training procedure, and second-stage bi-level optimization are kept fixed; only the last-layer prior is changed. We report performance both after Stage-1 probabilistic pretraining (Stage-1 model) and after the second-stage fine-tuning guided by the corresponding uncertainty (Stage-2 model). All variants are run with three random seeds, and we report mean and standard deviation.

Table 14: Effect of the sparsity prior on Waterbirds. Comparison between a standard Gaussian prior and the non-negative, sparsity-inducing Gamma prior used in our Bayesian last layer. We report worst-group accuracy (WGA) and mean accuracy (ACC) after Stage-1 probabilistic pretraining (Stage-1 model) and after Stage-2 uncertainty-guided fine-tuning (Stage-2 model). Results are mean $\pm$ standard deviation over three runs.

| Prior | Stage-1 WGA (%) | Stage-1 ACC (%) | Stage-2 WGA (%) | Stage-2 ACC (%) |
|---|---|---|---|---|
| Gaussian Prior | $48.87 \pm 0.5$ | $92.95 \pm 1.4$ | $7.32 \pm 0.1$ | $76.75 \pm 0.4$ |
| **Gamma Prior (Ours)** | $\mathbf{72.4 \pm 0.9}$ | $91.2 \pm 1.3$ | $\mathbf{92.8 \pm 1.5}$ | $\mathbf{95.1 \pm 0.4}$ |

As shown in Table 14, during probabilistic pretraining the Gamma prior already yields substantially higher worst-group accuracy than the Gaussian prior, despite having comparable mean accuracy. More importantly, under the same second-stage optimization, the Gaussian-prior model degrades to very low WGA and mean accuracy, whereas the Gamma-prior model improves both metrics markedly. This indicates that uncertainty-guided fine-tuning is substantially more effective when coupled with the non-negative, sparsity-inducing Gamma prior, while replacing it with a dense Gaussian prior leads to noticeably weaker robustness.

To further understand this difference, we also measure how much the last-layer parameters change before and after the second-stage fine-tuning. On Waterbirds, we compute the proportion of last-layer weights whose absolute change exceeds a threshold $\delta$, with $\delta \in \{0, 10^{-4}, 10^{-3}, 10^{-2}\}$. The results are summarized in Table 15.

Table 15: Proportion (%) of last-layer weights on Waterbirds whose absolute change exceeds a threshold $\delta$ during Stage-2 fine-tuning, for different priors. Lower values indicate more localized parameter updates.

| Prior | $\delta = 0$ | $\delta = 10^{-4}$ | $\delta = 10^{-3}$ | $\delta = 10^{-2}$ |
|---|---|---|---|---|
| Gaussian Prior | 100 | 99.76 | 97.93 | 79.24 |
| **Gamma Prior (Ours)** | **58.39** | **58.22** | **56.39** | **28.70** |

Under the Gaussian prior, almost all last-layer parameters undergo substantial updates across all thresholds, whereas under the Gamma prior the updates are concentrated on a much smaller subset of weights. In other words, the Gamma prior not only yields better worst-group performance, but also leads to more localized and sparse parameter changes in the second stage, which is aligned with the design goal of performing small, controlled edits on top of a fixed backbone. Overall, these ablations indicate that the non-negative, sparsity-inducing Gamma prior is not a replaceable implementation detail, but plays a crucial role in enabling both robustness gains and localized parameter editing in the proposed framework.

### D.3 ABLATION: SENSITIVITY TO $p$-VALUE THRESHOLD

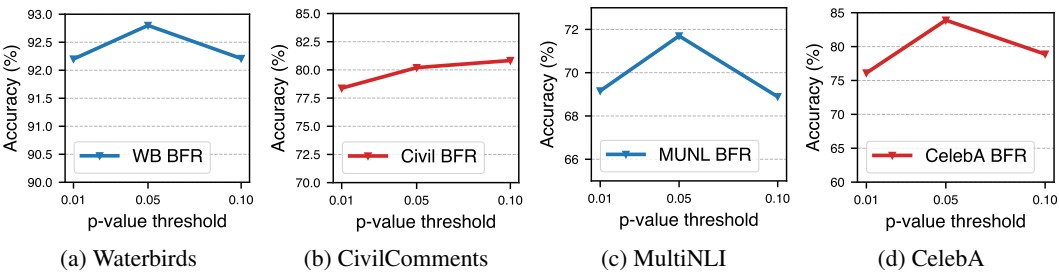

(a) Waterbirds     (b) CivilComments     (c) MultiNLI     (d) CelebA

Figure 6: Test Worst Group Accuracy of BFR for different uncertainty threshold (chosen from 0.01, 0.05 and 0.1) for different dataset.

Figure 6 depicts how the test worst-group accuracy of the BFR method on the different datasets varies with different uncertainty thresholds. The horizontal axis lists three p-value thresholds (0.01,

0.05,0.10), while the vertical axis represents the worst group accuracy (WGA). The solid line with triangular markers indicates that BFR surpasses ERM across different $p$-value settings. On most datasets, performance peaks at $p = 0.05$; accordingly, we adopt 0.05 as the uncertainty threshold throughout this paper. We also conduct a more comprehensive **sensitivity analysis for a set of Hyper-parameters (including $p$-value) in Fig. 10.**

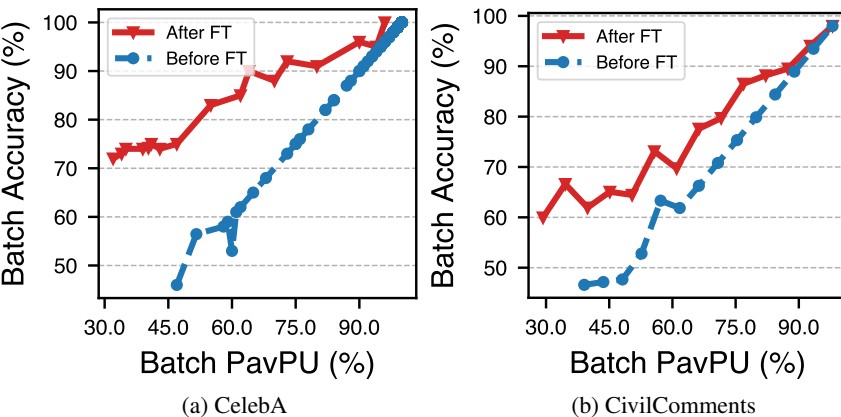

(a) CelebA  (b) CivilComments

Figure 8: PavPu-Accuracy curves for **(a)** CelebA and **(b)** CivilComments datasets. Red: after uncertainty-guided bi-level optimization; blue: before.

### D.4 ABLATION: RELATION BETWEEN UNCERTAINTY AND ACCURACY

Corresponding to Sec.4.2, we provide Uncertainty–Accuracy curves on two additional datasets under the same experimental setup. We observe a consistent pattern: after uncertainty-guided fine-tuning, the curves shift up and to the left—indicating higher accuracy at the same uncertainty level (and lower uncertainty at the same accuracy). This provides further evidence that uncertainty-guided fine-tuning improves model robustness.

### D.5 ABLATION: EFFECTIVENESS OF BI-LEVEL OPTIMIZATION

Table 16: Hyperparameter grid used for the **BiLevel (BFR)** method in the sensitivity analysis of Fig.10. Here, *inner learning rate* denotes the step size for the inner maximization in the bi-level optimization, and $p$-value is the uncertainty threshold.

| Hyperparameter | Set of values |
| --- | --- |
| inner learning rate (BiLevel) | $\{1\times10^{-4}, 5\times10^{-4}, 1\times10^{-3}\}$ |
| $p$-value (BFR) | $\{0.01, 0.05, 0.1\}$ |
| random seed | $\{0, 1, 2\}$ |

Table 17: Hyperparameter grid used for the **Direct-U** baseline in the sensitivity analysis of Fig.10. Here, *learning rate* refers to the standard optimizer step size in the one-stage weighted ERM fine-tuning.

| Hyperparameter | Set of values |
| --- | --- |
| learning rate (Direct-U) | $\{1\times10^{-4}, 5\times10^{-4}, 1\times10^{-3}\}$ |
| random seed | $\{0, 1, 2\}$ |

For the Waterbirds dataset, we conduct a hyperparameter sensitivity analysis by performing a grid search over the configurations listed in Table 16 and Table 17, yielding 27 runs for **BiLevel** and 9 runs for **Direct-U**, respectively (see Fig. 10). For the second stage of BFR, we denote our full method, which employs uncertainty-guided bi-level optimization, as **BiLevel**. The variant without

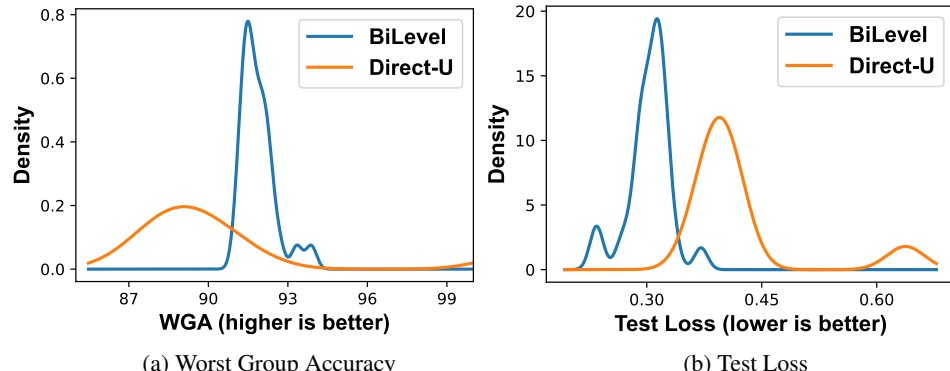

(a) Worst Group Accuracy        (b) Test Loss

Figure 10: Sensitivity of training schemes over hyperparameters sampled from the grids in Table 16 (BiLevel) and Table 17 (Direct-U), respectively. Kernel-density distributions over (a) worst–group accuracy (higher is better) and (b) test loss (lower is better) for BiLevel vs. Direct-U, aggregated across a grid of learning rates, uncertainty thresholds, and random seeds. BiLevel yields sharper, shifted densities—toward higher WGA and lower loss—indicating better performance and markedly reduced hyperparameter sensitivity; Direct-U exhibits broader/heavier tails, reflecting less stable convergence.

bi-level optimization is denoted **Direct-U**: after the probabilistic pre-training stage, Direct-U computes the instance-level predictive uncertainty $U(x)$ for each validation sample and uses $U(x)$ as a sample weight $w(x)$; in the subsequent fine-tuning stage, it optimizes a weighted empirical risk

$$\mathcal{L}_{\text{Direct-U}}(\theta) = \mathbb{E}_x\big[w(x)\,\ell(f_\theta(x))\big],$$

i.e., higher-uncertainty samples receive larger weights in the loss. As shown in Fig. 10, Direct-U is considerably more sensitive to the choice of hyperparameters, whereas BiLevel consistently attains better worst-group accuracy and lower test loss across the examined configurations.

## D.6 COMPLEXITY ANALYSIS

To assess the practical computational cost of different methods, we compare several representative group-robustness approaches on the Waterbirds dataset, including DFR (Kirichenko et al., 2022), BAM (Li et al., 2023), GSR (Qiao et al., 2025), and our proposed BFR. Table 18 reports both inference and overall training overhead: GFLOPs denotes the number of floating-point operations for a single forward pass, P50/P90 are the median and 90th-percentile inference latency on a single GPU, Memory is the peak GPU memory consumption during inference, and total training time is the overall wall-clock time required to train the model (including both stages). We observe that BFR closely matches other methods in terms of GFLOPs, inference latency, and memory usage, and its total training time is on the same order as DFR and GSR—only slightly higher than DFR and substantially lower than BAM. This indicates that, even with the additional uncertainty-guided bi-level optimization, BFR maintains training and inference costs comparable to existing two-stage methods, which is consistent with our theoretical complexity analysis in Sec.3.4.

## D.7 VISUALIZATION

Additional visualization results. We selected the top samples with the highest and lowest uncertainty, respectively, and visualized their most highly activated features, as shown in Fig .11. The second row: heatmaps for high-uncertainty samples often fall on background or spurious regions, whereas those for low-uncertainty samples primarily focus on key parts of the target. The third row: the attention of high-uncertainty samples is corrected, with heatmaps converging on semantically relevant objects; low-uncertainty samples further contract to a more concentrated core area. Overall, these results demonstrate that uncertainty can effectively identify and drive the correction of spurious features, thereby increasing the model's emphasis on truly discriminative cues and improving group robustness.

Table 18: Overall training and inference cost on Waterbirds for two-stage methods, where a base model is first trained and then fine-tuned with a method-specific second stage. We report per-image GFLOPs, median and 90th percentile latency (P50/P90), peak GPU memory usage, and the total wall-clock training time. **Lower values indicate better efficiency**. BFR exhibits comparable computational cost to other baselines.

| Waterbirds | GFLOPs | P50 (ms) | P90 (ms) | Memory (MiB) | total training time (min) |
|---|---|---|---|---|---|
| DFR | 0.047 | **4.03** | 4.55 | **232.57** | 14.60 |
| BAM | 0.047 | 4.70 | 4.78 | 232.66 | 57.32 |
| GSR | 0.047 | 4.82 | 5.59 | **232.57** | 17.34 |
| BFR | 0.047 | 4.04 | **4.24** | 232.58 | 16.33 |

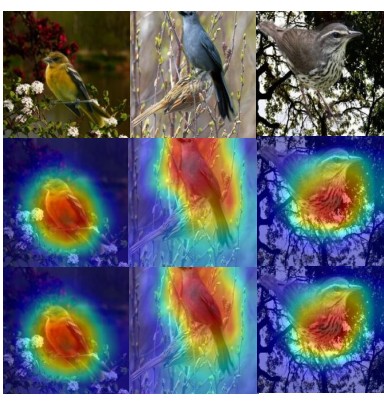

Figure 11: GradCAM (Selvaraju et al., 2017) visulizations for BFR on Waterbirds dataset. We selected the top samples from both the high-uncertainty and low-uncertainty (certain) groups. The second row shows the visualizations after the pre-training stage, and the third row shows the results after uncertainty-guided fine-tuning. Spuriously correlated samples are identified by high uncertainty and are corrected after fine-tuning; samples with low uncertainty consistently focus on the core features and exhibit an even more concentrated pattern after fine-tuning.

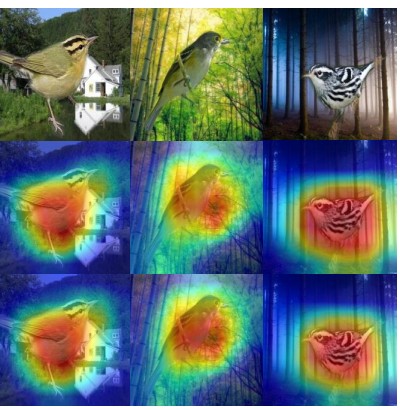

Figure 12: GradCAM (Selvaraju et al., 2017) visulizations for BFR on Waterbirds dataset.

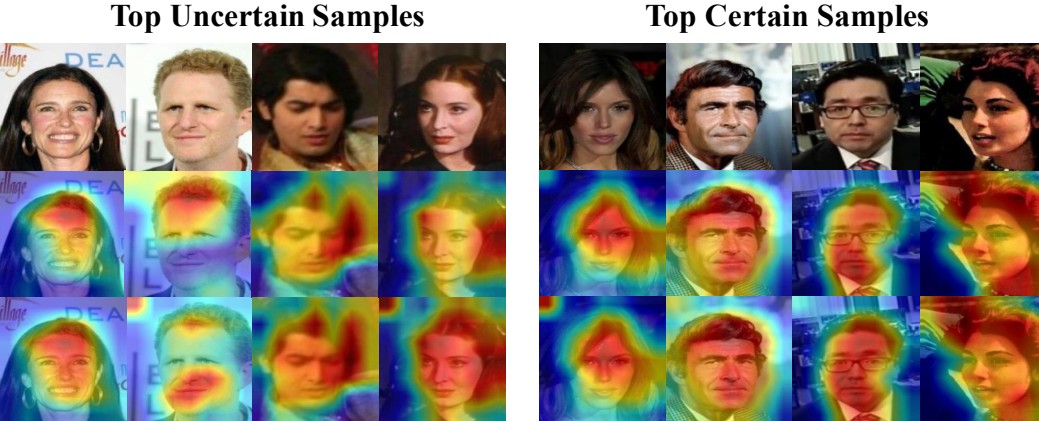

Figure 13: GradCAM (Selvaraju et al., 2017) visulizations for BFR on CelebA dataset.

