# OpenReview forum: "Turning Uncertainty into Control: Bi-level Training with Editable Bayesian Layers"
_ICLR.cc/2026/Conference — ICLR 2026 Conference Withdrawn Submission_

### Official Review · Reviewer_RT8U · 2025-10-31

**Soundness:** 2
**Presentation:** 3
**Contribution:** 2
**Rating:** 4
**Confidence:** 4

**Summary:**

The authors present Bayesian Feature Reweighting, a two-step procedure which first uses probabilistic pre-training with a Bayesian last layer to then perform uncertainty-guided optimization. They show that their method achieves competitive accuracy and worst-group accuracy across a diverse set of benchmarks.

**Strengths:**

- The authors do a thorough job of benchmarking their method against a large number of related methods on various spurious correlation benchmarks, and BFR has competitive results.
- The method is thoroughly explained, and each step has sufficient detail. The intuitions behind the method are presented well, and the code is also available for reproducibility.
- The visualizations and ablations provided are interesting, and Figure 4 demonstrates how the model is able to shift from spurious correlation to core features for high-uncertainty samples.

**Weaknesses:**

- I do not think that the method is well-motivated, although I may have missed this in the paper (see next point). Here is my understanding of the main claims of the paper, and why I don't think the evidence is sufficient to support these claims.
  1. **The sparsity prior is important**: The prior over $\theta$ and $\Phi$ is fixed to $Gamma(1, 1)$ throughout the paper, and there are no ablations with other choices of priors. It's unclear to me what the benefits of this prior are.
  2. **BLL is a good method of measuring uncertainty**: BLL is the only probabilistic method which is tested, and the authors do not ablate with other low-cost methods like Dropout. I would also be interested in seeing an ablation where epistemic uncertainty is not considered, and the bi-level optimization is performed directly on the original's model's aleatoric uncertainty, since that would demonstrate the importance of the probabilistic pre-training component of the method.
  3. **Uncertainty-guided bi-level optimization outperforms other instance-level uncertainty methods**: While the proposed method is intuitive, there seem to be simpler methods to balancing high-uncertainty and low-uncertainty loss, such as minimizing a weighted sum of the two groups. It would be helpful to see how this minmax problem is beneficial. I see an ablation with "Direct-U" in the appendix (not referenced in the main text), but I don't understand what this weighting is, or why there would be an inner learning rate for these problems.
- The analysis of the empirical results is weak, and makes the result difficult to interpret. The methods which BFR is compared against are not explained. Even if there is insufficient space to discuss all of the results in detail, it would be helpful to directly compare the differences between BFR and a few related works so I can understand why BFR seems to be performing well. Instead, the paper concludes (L484) "Additional ablations isolate the contributions of each uncertainty component and of the retraining strategy", but I don't know which experiments this refers to.

**Questions:**

- Could you elaborate on what ERM-BFR means? This seems like an important ablation, but the only explanation I could find was L355: "We replace the last layer of the base ERM models (Vapnik, 1999) for the base model training". How does this differ from BFR?
- Do you have runtimes for your method? I see that there's a complexity analysis, but it's difficult to understand how significant the big O terms are in practice.

---

> ### Author Response · Authors · 2025-11-19
>
> We thank the reviewer for the careful reading and for the constructive questions. Our detailed responses are as follows:
>
> **Summary**:
>
> - We added new ablations on ***prior choices*** (Gamma vs. Gaussian) and on ***uncertainty estimators***/types (entropy-based, dropout-based, aleatoric, epistemic, and full predictive uncertainty).
>
> - We clarified ***the Direct-U baseline as a weighted-sum variant suggested by the reviewer***, and used hyperparameter sensitivity experiments to show the benefits of the min–max bi-level formulation over this simpler alternative.
>
> - We strengthened the empirical and complexity analysis by better categorizing baselines, ***expanding the discussion of Tables 1 and 2 on Sec 4.1***, and adding ***wall-clock runtime measurements*** to make the computational overhead of BFR more explicit.
>
> ---
>
> > **Q1**: Could you elaborate on what ERM-BFR means?
>
> Regarding the concern on “ERM-BFR”, we are sorry about this misunderstanding, this is indeed central to our ablation design. Our training procedure can be viewed as consisting of two stages:
>
> (1) **Probabilistic pretraining phase.** In this stage, we start from a standard ERM network architecture and probabilize the output layer. We then pretrain the resulting network on the training set using the ELBO objective of the probabilistic model. **The intermediate model obtained at the end of this phase is what we refer to as ERM-BFR.** Its main difference from standard ERM is that the decision layer is a BLL with sparsity-inducing priors, which yields better-calibrated predictive uncertainty. In Appendix D.2 (Table 10), we provide a comparison between ERM and ERM-BFR in terms of calibration metrics such as ECE, showing that ERM-BFR substantially improves calibration with almost no loss in accuracy, thereby providing a more reliable basis for subsequent uncertainty-guided steps.
>
> (2) **Uncertainty-guided bi-level optimization phase.** Next, we initialize from ERM-BFR and use its instance-level predictive uncertainty on the validation set to perform uncertainty-guided bi-level optimization on the BLL only. **The model obtained after this second-stage fine-tuning is the final BFR reported in the paper.**
>
> **In short, ERM-BFR can be viewed as ‘BFR without the uncertainty-guided bi-level retraining stage‘**. We clarified this definition in the revised manuscript in **line 406**. We hope this clarification makes the role of ERM-BFR and its difference from BFR more transparent.

---

> > ### Author Response · Authors · 2025-11-19
> >
> > > **W1.1**:  The sparsity prior is important. The prior over $\theta$ and $\Phi$ is fixed to $Gamma(1, 1)$ throughout the paper, and there are no ablations with other choices of priors. It's unclear to me what the benefits of this prior are.
> >
> > Thank you for pointing out that. We agree that the sparsity-inducing prior is a key component of our method. Accordingly, in the revised manuscript **(Appendix D.1 Table 9 and Table 10)** we have added an ablation study comparing our prior against alternative choices to more clearly illustrate its concrete benefits.
> >
> > Specifically, on Waterbirds dataset we keep the backbone, training procedure, and second-stage bi-level optimization entirely unchanged, and only replace the Bayesian last-layer prior from the Gamma prior used in the paper (non-negative and sparsity-inducing) with a standard Gaussian prior $ \mathcal{N}(0,1) $; we denote this variant as **Gaussian Prior**. We then report performance both after Stage-1 probabilistic pretraining (Stage-1 model) and after the second-stage fine-tuning guided by the corresponding uncertainty (Stage-2 model). For a fair comparison, all variants are run with three random seeds and we report the mean and standard deviation. The results are as follows:
> >
> >
> > |     Stage-1 model    | wga (\%)          | acc (\%)          |
> > |----------------------|-------------------|-------------------|
> > |    Gaussian Prior    | 48.87 $\pm$ 0.5   | 92.95 $\pm$ 1.4   |
> > | Gamma Prior (ERM-BFR)| **72.4 $\pm$ 0.9**    | 91.2 $\pm$ 1.3    |
> >
> > |   Stage-2 model  | wga (\%)          | acc (\%)          |
> > |------------------|-------------------|-------------------|
> > |   Gaussian Prior | 7.32 $\pm$ 0.1    | 76.75 $\pm$ 0.4   |
> > | Gamma Prior (BFR)| **92.8 $\pm$ 1.5**    | 95.1 $\pm$ 0.4    |
> >
> >
> > In the probabilistic pretraining stage, the model with the Gaussian prior attains a worst-group accuracy of only 48.87, whereas the Gamma prior already yields a substantially higher WGA, despite having comparable mean accuracy. More importantly, under the same second-stage fine-tuning procedure, the Gaussian-prior model degrades to 7.32 in WGA and 76.75 in mean accuracy, while the Gamma-prior model improves to 92.8 and 95.1, respectively.
> > These results indicate that uncertainty-guided fine-tuning is substantially more effective when coupled with the non-negative, sparsity-inducing Gamma prior: with all other components fixed, the Gamma prior consistently improves WGA while maintaining high mean accuracy, whereas replacing it with a dense Gaussian prior leads to noticeably weaker robustness.
> >
> >
> > | waterbirds | 0      | 1e-4    | 1e-3    | 1e-2    |
> > |------------|--------|---------|---------|---------|
> > | Gaussian   | 100    | 99.76   | 97.93   | 79.24   |
> > | Ours       | **58.39** | **58.22** | **56.39** | **28.70** |
> >
> > To further understand this discrepancy, we also measure how much the last-layer parameters change before and after the second-stage fine-tuning. On Waterbirds, we compute the proportion of last-layer weights whose absolute change exceeds a threshold $\delta$, with $\delta \in {0, 10^{-4}, 10^{-3}, 10^{-2}}$. The results show that under the Gaussian prior, almost all last-layer parameters undergo substantial updates, whereas under the Gamma prior the updates are concentrated on a much smaller subset of weights. In other words, the Gamma prior not only yields better worst-group performance, but also leads to more localized and sparse parameter changes in the second stage, which is aligned with our design goal of performing small, controlled edits on top of a fixed backbone.
> >
> > Taken together, this ablation study indicates that the non-negative, sparsity-inducing Gamma prior is not merely an implementation detail, but plays an important role in enabling both robustness gains and localized parameter editing. When we replace it with a structurally simpler Gaussian prior that lacks sparsity and non-negativity, the overall performance degrades substantially. We will include these ablation results and the corresponding discussion **in the appendix D.2 of the revised manuscript to more systematically present the benefits of this prior**.

---

> > > ### Author Response · Authors · 2025-11-19
> > >
> > > > **W1.2**: BLL is the only probabilistic method which is tested, and the authors do not ablate with other low-cost methods like Dropout. Ablation with other Uncertainty estimates method, and ablations on uncertainty type. Since that would demonstrate the importance of the probabilistic pre-training component of the method.
> > >
> > > Thank you for the helpful suggestion. We fully agree that, to demonstrate the importance of the probabilistic pre-training component, it is necessary both to compare against alternative uncertainty estimation methods and to perform ablations over different types of uncertainty within our own framework. Accordingly, we have added two sets of experiments in the revised manuscript. **Overall, our experiments show that (i) entropy- and dropout-based proxies are clearly weaker than BFR in worst-group accuracy and induce almost global last-layer updates, and (ii) within BFR, using full predictive uncertainty yields the largest robustness gains over using aleatoric or epistemic uncertainty alone.**
> > >
> > > (1) First, to address the question of whether BLL could be replaced by alternative proxies such as Dropout, we keep the ERM backbone and the subsequent bi-level optimization procedure entirely unchanged, and vary only the uncertainty signal used in the second stage. We consider three options: (i) a confidence-based proxy using the entropy of ERM outputs (Entropy-based), (ii) a Dropout-based proxy using the variance from MC Dropout sampling, and (iii) our proposed BFR that uses the predictive uncertainty from the Bayesian last layer. On Waterbirds and CelebA, we evaluate worst-group accuracy (WGA) and mean accuracy. The results are as follows:
> > >
> > >
> > > | Uncertainty Proxy    | Waterbirds WGA      | Waterbirds Mean ACC | CelebA WGA         | CelebA Mean ACC     |
> > > |----------------------|----------------------|-----------------------|---------------------|-----------------------|
> > > | Entropy-based        | 88.2 ± 1.1           | 96.2 ± 0.5            | 76.1 ± 0.9          | 92.0 ± 0.4            |
> > > | Dropout-based        | 87.2 ± 0.8           | 95.9 ± 0.5            | 68.3 ± 1.2          | 93.0 ± 0.6            |
> > > | **BFR (Ours)**       | **92.8 ± 1.5**       | **95.1 ± 0.4**        | **83.9 ± 1.0**      | **91.2 ± 0.7**        |
> > >
> > > The results show that, the WGA achieved by the entropy-based and dropout-based proxies is consistently and substantially lower than that of BFR on both datasets, while their mean accuracy remains at a similar level to BFR. In other words, these proxy methods are able to maintain overall accuracy but are clearly less effective than BFR with a Bayesian last layer in identifying bias-conflicting samples and improving worst-group accuracy.
> > >
> > > To further analyze the differences between these proxies at the level of parameter editing, we follow the metric used in Table 3 of the main paper and measure, on Waterbirds, the proportion of last-layer weights whose absolute change exceeds a threshold $\delta$, with $\delta \in {0, 10^{-4}, 10^{-3}, 10^{-2}}$. The results are as follows:
> > >
> > >
> > > | waterbirds     | 0       | 1e-4    | 1e-3    | 1e-2    |
> > > |----------------|---------|---------|---------|---------|
> > > | Entropy-based  | 100     | 99.82   | 96.24   | 64.54   |
> > > | Dropout-based  | 100     | 99.85   | 97.68   | 75.81   |
> > > | **BFR (Ours)** | **58.39** | **58.22** | **56.39** | **28.70** |
> > >
> > > In the entropy-based and dropout-based cases, the proportion of weights whose change exceeds each threshold is essentially close to 100\%, indicating that the second-stage fine-tuning induces substantial updates to almost all last-layer parameters. In contrast, for BFR this proportion is markedly lower (for example, at $\delta = 10^{-2}$, only about 28.7\% of the weights undergo a noticeable change), suggesting that the updates are much more localized and sparse. Taken together with the performance results, this comparison indicates that, within our framework, simple entropy- or dropout-based uncertainty is not only less effective in achieving comparable robustness gains, but also does not support the same level of localized and controllable parameter editing as BLL.

---

> ### Author Response · Authors · 2025-11-19
>
> >  **W1.2 continue**:
>
> (2) Second, to more directly characterize whether probabilistic pre-training itself is necessary and how different types of uncertainty contribute, we conducted an additional ablation in which we keep the overall framework fixed and vary only the uncertainty signal used for the second-stage bi-level optimization.
> Specifically, we compare (i) plain ERM without probabilistic pre-training, (ii) using only aleatoric uncertainty $\theta$, (iii) using only epistemic uncertainty $\Phi$, and (iv) using the full predictive uncertainty from BFR. We reported mean $\pm$ std over 3 random runs, other settings follow Table 1 in the main text. The results on four datasets are summarized as follows:
>
> | Uncertainty        | Waterbirds WGA       | Waterbirds Mean      | CelebA WGA           | CelebA Mean          | MNLI WGA             | MNLI Mean            | Civil WGA            | Civil Mean           |
> |--------------------|----------------------|------------------------|------------------------|------------------------|------------------------|------------------------|------------------------|------------------------|
> | ERM                | 71.9 ± 1.5          | 91.4 ± 1.7            | 45.1 ± 0.8            | 95.1 ± 0.4            | 59.2 ± 0.3            | 81.9 ± 0.1            | 54.6 ± 0.6            | 91.4 ± 0.1            |
> | Aleatoric ($\theta$)      | 85.98 ± 0.6          | 95.58 ± 0.3           | 78.33 ± 0.4           | 92.26 ± 0.2           | 70.12 ± 0.8           | 80.60 ± 0.2           | 67.91 ± 0.7           | 91.41 ± 0.3           |
> | Epistemic ($\Phi$)      | 87.38 ± 0.9          | 95.65 ± 0.3           | 68.89 ± 0.6           | 92.93 ± 0.3           | 70.91 ± 0.9           | 80.55 ± 0.3           | 73.64 ± 0.8           | 90.32 ± 0.2           |
> | **Predictive (BFR)** | **92.8 ± 1.5**      | 95.1 ± 0.4            | **83.9 ± 1.0**        | 91.2 ± 0.7            | **73.8 ± 0.9**        | 80.8 ± 0.5            | **80.5 ± 1.6**        | 88.2 ± 0.3            |
>
> We observe three trends. First, compared to ERM, using either aleatoric or epistemic uncertainty from the Bayesian last layer already yields a substantial gain in worst-group accuracy across all datasets, indicating that a purely deterministic ERM model does not provide a sufficiently informative uncertainty signal for guiding robust training. Second, while aleatoric and epistemic uncertainty individually help, the full predictive uncertainty derived from the joint posterior $(\theta, \Phi)$ consistently achieves the highest worst-group accuracy on all benchmarks (improvements of roughly 5–7 percentage points over the single-component variants). This supports our claim that probabilistic pre-training is not a cosmetic choice: explicitly modeling and disentangling aleatoric and epistemic uncertainty is necessary in order to construct a strong predictive uncertainty signal for Stage 2. Third, these robustness gains are obtained with competitive or modestly traded-off mean accuracy, suggesting that the probabilistic pre-training plus uncertainty-guided fine-tuning leads to targeted corrections rather than indiscriminate overfitting to minority examples.
>
> Taken together, these two ablations indicates that, on the one hand, under the same backbone, directly replacing BLL with uncertainty estimates based on Dropout or entropy leads to noticeably worse worst-group accuracy and much less localized parameter updates compared to BFR. On the other hand, the full predictive uncertainty enabled by the Bayesian last layer is crucial for realizing the benefits of our framework, thereby highlighting the importance of the probabilistic pre-training component. We thank the reviewer again for this helpful suggestion. **We will incorporate the above comparative experiments and discussion into the revised manuscript (appendix D.1 and D.2) to more systematically quantify and demonstrate the role of probabilistic pre-training.**

---

> ### Author Response · Authors · 2025-11-19
>
> > **W1.3**: Uncertainty-guided bi-level optimization ... but I don't understand what this weighting is, or why there would be an inner learning rate for these problems.
>
> We thank the reviewer for raising this point. The suggested idea of “minimizing a weighted sum of the two groups” is indeed a very natural baseline, and it directly motivated the Direct-U ablation we included in the appendix D.5. Specifically, after the probabilistic pre-training stage, Direct-U computes the instance-level predictive uncertainty $U(x)$ for each validation sample and uses $U(x)$ as a sample weight $w(x)$. In the second-stage fine-tuning, it then optimizes a weighted empirical risk:
>
> $$
> L_{Direct-U}(\theta) = 𝔼_{x}[w(x)l(f_{\theta}(x))],
> $$
>
> **which is precisely a concrete realization of the reviewer’s suggestion of “minimizing a weighted sum of the two groups”: high-uncertainty samples receive larger weights, and low-uncertainty ones are down-weighted.**
>
> Compared with this single-stage, fixed-weight weighted-sum scheme, our uncertainty-guided bi-level optimization treats the sample weights as an inner variable and approximately solves an inner maximization at each step, so that the weights adapt to the current model and focus more on high-risk samples. In Fig. 9, we perform a hyperparameter sensitivity analysis for both methods under the same hyperparameter grid, and we observe that across all four datasets, the bi-level formulation consistently achieves higher worst-group accuracy and exhibits more stable behavior with respect to hyperparameter changes, indicating that the min–max structure provides additional benefits beyond a simple weighted-sum objective.
>
> Regarding the question about why there is an inner learning rate, we apologize for the confusion. In Fig. 9, we used a shared hyperparameter grid for BFR and Direct-U (primarily designed around BFR) for fairness. To clarify: the inner learning rate is an algorithmic hyperparameter introduced solely to solve the inner maximization problem in the bi-level method. Direct-U does not involve any inner optimization; it is implemented as a single-stage weighted ERM fine-tuning and therefore only uses a standard outer learning rate, without any inner learning rate. We acknowledge that the description of Direct-U in the appendix was not sufficiently clear and may have caused misunderstanding. In the revision, **we will (i) explicitly reference the Direct-U ablation from Sec 3.5 and (ii) rephrase the appendix D.5 to clearly distinguish these two setups.**

---

> ### Author Response · Authors · 2025-11-19
>
> > **W2**: The analysis of the empirical results is weak, and makes the result difficult to interpret. ...the paper concludes (L484)... but I don't know which experiments this refers to.
>
> We sincerely thank the reviewer for this comment. We recognize that the presentation of the empirical results in the original version could be further clarified and strengthened, particularly in terms of highlighting the differences between each baseline and BFR, and in making the references to the ablation studies in the appendix more transparent. Following your suggestions, we have made the following revisions:
>
> 1. In Table 1 of the main text, **we added an additional column that provides a more detailed categorization of the compared methods. This corresponds to the description in **Baseline paragraph in Sec. 4** in the revised manuscript and serves as an explicit grouping of the baselines.** We also briefly explain the design goals of each category and clarify where BFR fits within this spectrum (i.e., using predictive uncertainty from a Bayesian last layer to approximately minimize worst-group risk without access to group labels).
>
> 2. **In "Improvement in Group Roubustness" paragraph in Sec 4.1**, we ***refined the analysis of both Table 1 and Table 2*** to make the empirical conclusions easier to interpret (***highlighted in blue***). For Table 1, we now explicitly compare BFR against the three categories of methods and emphasize that, under the fully group-annotation-free setting, BFR achieves the highest worst-group accuracy across all four datasets, while on benchmarks such as Waterbirds it already matches or even slightly surpasses group-aware methods like Group DRO and DFR. At the same time, BFR matches or outperforms meta-training–based methods such as MAPLE and GSR, which rely on partial group annotations, on most tasks. For Table 2, we provide a more detailed discussion of the ImageNet-9 → ImageNet-A distribution shift experiment, highlighting that BFR maintains strong accuracy on ImageNet-9 while achieving the best top-1 accuracy on the more challenging ImageNet-A among all baselines, and significantly reducing the performance gap between ID and OOD data. These additional explanations help readers more directly understand BFR’s advantages over different types of baselines and its behavior under distribution shift.
>
> Regarding your concern that the reference to “Additional ablations …” at line 484 was unclear, **we now explicitly refer to Appendix D in Section 4.1 of the main text and briefly summarize the main conclusions of these ablations in line **472-476**.** In the revised manuscript, we also add an overview paragraph at the beginning of Appendix D, clearly stating that we systematically analyze: (i) predictive uncertainty versus its aleatoric and epistemic components, (ii) the effect of the significance-test (p)-value threshold on performance, (iii) the separate contributions of the probabilistic pretraining stage and the bi-level optimization stage, and (iv) the additional computational overhead incurred by BFR. At the same time, we concisely summarize the key findings of these ablations at line **472-476** to make their role in the overall empirical narrative more apparent.
>
> We hope these revisions help address your concerns about the empirical analysis and make the relationship between BFR’s results and existing methods clearer and easier to interpret.
>
> ---
>
> > **Q2**: Do you have runtimes for your method?
>
> Thank you for pointing out that Big-O complexity alone makes it difficult to gauge the actual computational overhead. As clarified in our global response, we have supplemented the revised manuscript with wall-clock runtime measurements and explicitly referenced them in Sec. 3.4. Specifically, on the Waterbirds dataset, we conducted a unified evaluation of GFLOPs, inference latency, inference-time memory consumption, and total training time for DFR, BAM, GSR, and BFR (Appendix D.6, Table 18). The results show that the total training time of BFR is 16.33 minutes, compared to 14.60 and 17.34 minutes for DFR and GSR, respectively, and substantially lower than BAM’s 57.32 minutes; meanwhile, all four methods exhibit nearly identical GFLOPs, P50/P90 inference latency, and memory usage. These findings indicate that the additional overhead introduced by our method during actual execution is limited and comparable. Please refer to the global response and Table 18 in Appendix D.6 for the complete numerical results and analysis.

---

### Official Review · Reviewer_jkMj · 2025-11-01

**Soundness:** 2
**Presentation:** 2
**Contribution:** 2
**Rating:** 4
**Confidence:** 3

**Summary:**

This paper introduces Bayesian Feature Reweighting (BFR), a framework that uses calibrated predictive uncertainty as a control signal for model training. It combines a Bayesian Last Layer with sparsity-inducing priors and a bi-level optimization procedure that reweights samples based on uncertainty, promoting robust and localized parameter updates. BFR is tested on several vision and NLP benchmarks and improves worst-group accuracy and out-of-distribution robustness without requiring group labels.

**Strengths:**

1. **Motivation:** The paper moves beyond using uncertainty only for post-hoc evaluation and instead leverages it as a training signal to guide model updates, and allows robust fine-tuning. This provides an effective approach to improving robustness and editability.

2. **Evaluation:** The method is evaluated across diverse vision and NLP benchmarks, demonstrating consistent gains in worst-group accuracy and robustness over strong baselines.

3. **Visualization:** Results in Fig 4 shows an interesting observation that how BFR produces more interpretable model changes, aligning with its goal of controlled and explainable fine-tuning.

4. **Theoretical and complexity analysis:** This work also introduces a theoretical and complexity analysis, increase the soundness of the proposed method.

**Weaknesses:**

1. **Motivation:**
The method performs well without group annotations but remains weaker than baselines that use them. The paper does not clearly justify why annotation-free settings are particularly important or how frequently they occur in practice. A clearer motivation and discussion of use cases would strengthen the contribution.

2. **Scalability:** Although the authors claim negligible computational overhead, the probabilistic formulation and bi-level optimization still introduce nontrivial costs. It remains unclear how the method scales to larger models and datasets, what the typical training time and computational resources are, and whether these factors limit broader applicability.

3. **Last Layer fine-tuning:** The Bayesian Last Layer is efficient but inherently limited in scope since only the final layer is adapted. This may restrict representational flexibility and the model’s ability to capture deeper interactions compared to methods that fine-tune intermediate representations. A discussion or comparison illustrating this trade-off would be valuable.

**Questions:**

1. Could the authors elaborate on the practical importance of group-annotation-free settings? In which real-world applications is this assumption most relevant? Clarifying why this scenario deserves particular attention would strengthen the paper’s motivation, especially since methods with group labels still outperform BFR.

2. While the Bayesian Last Layer is presented as having negligible overhead, it only adapts the top layer during fine-tuning. Could the authors discuss whether this limits representational flexibility compared to fine-tuning other layers?
Computational scalability.

3. It would be helpful to include or discuss results on how sensitive BFR’s performance is to hyperparameters. Which components contribute most to the observed robustness gains?

---

> ### Author Response · Authors · 2025-11-19
>
> We sincerely thank the reviewer for the thoughtful comments, we first summarize our feedbacks as follows:
>
> **Summary**:
>
> - We further clarify the motivation and practical importance of the ***group-annotation-free setting***, and position our method more explicitly relative to existing work in this space.
>
> - We augment the paper with new ***computational cost experiments***, providing quantitative evidence for the training/inference efficiency and scalability of BFR.
>
> - We discuss the ***trade-off between “last-layer-only” fine-tuning and representational flexibility***, and show via ablations that gradually unfreezing more layers degrades robustness, indicating that restricting Stage 2 to the Bayesian last layer offers a favorable robustness–efficiency balance.
>
> - We systematically analyze BFR’s sensitivity to hyperparameters and the relative contributions of its components ***(Ablations on uncertainty type, probabilistic pretraining stage, and bi-level optimization stage)***, and we add the corresponding ablation results to ***Appendix D***.
>
> ---
>
> > **W1 and Q1**: Motivation. The method performs well without group annotations but remains weaker than baselines that use them...would strengthen the contribution.
>
> Thank you for your question. Regarding the motivation and for the attention to our results under the group-annotation-free setting, we would like to further clarify the motivation behind this setting and how it relates to practical applications.
>
> First, following the reviewer’s suggestion, we will revise the introduction and motivation to more explicitly articulate why annotation-free scenarios are prevalent in practice. Unlike class labels, which simply specify the semantic category of an instance, group annotations require assigning each sample to a finer-grained subpopulation or environmental condition. For example, in a standard cat-vs-dog classification task, the class label only indicates whether an image contains a cat or a dog, whereas a group annotation additionally identifies the underlying subgroup, such as “dog on grass’’ versus “dog indoors’’ (class × background), we further provide a dataset example in Fig. 5 in Appendix C.4.1. Such group labels typically require domain expertise or auxiliary metadata to acquire. In many real-world group-robust applications, such as weather and road conditions in autonomous driving, or sensitive attributes in fairness settings, reliable group labels are difficult or even impossible to obtain [1]. Unlike class labels, group definitions often involve expert knowledge or privacy-sensitive information, making “sufficiently fine-grained and accurate’’ group annotation essentially non-scalable in most deployed systems [2].
>
> These challenges have motivated a growing line of work aiming to reduce or eliminate reliance on group annotations [1 - 5]. However, many of these approaches still require a non-trivial amount of group labels for model selection, or incur substantial computational cost due to repeated full-model retraining or explicit mining of minority subgroups. Our method differs from these approaches in two ways: (i) we retrain only the Bayesian Last Layer and incorporate a bi-level update to adjust feature weights, leading to lower computational overhead and easier integration into existing systems compared with methods requiring full-model retraining or additional auxiliary models; and (ii) under this more realistic constraint, we provide a group-annotation-free training pathway that relies solely on model-intrinsic uncertainty to identify and emphasize implicit minority subpopulations.
>
> Second, we agree with the reviewer that, under the current benchmark protocol, our annotation-free method performs slightly below baselines that have access to explicit group labels. We believe this reflects the supervision gap: group-aware methods are allowed to use group information during training or model selection, whereas our method operates without any form of group or pseudo-group supervision. Our goal is therefore not to surpass group-aware methods under stronger supervision, but to substantially improve group robustness under the same group-annotation-free conditions and to outperform existing group-annotation-free approaches. In the revised Table 1, we will more clearly differentiate methods according to whether group labels are visible during training or selection.
>
> Based on the your feedback, we will revise the introduction and motivation accordingly, and **will further highlight our differences and advantages relative to prior work in Appendix C.4.1**.

---

> > ### Author Response · Authors · 2025-11-19
> >
> > > **Scalability:** Although ... what the typical training time and computational resources are, and whether these factors limit broader applicability.
> >
> > We appreciate the reviewer’s concerns about scalability. In our global response, we provided additional discussion on computational overhead and scalability, and we have incorporated corresponding experiments into the revised manuscript (Sec. 3.4 and Appendix D.6). Structurally, the second stage of BFR performs uncertainty-guided bi-level updates only on the Bayesian last layer while keeping the backbone fixed. Its additional computational cost therefore scales linearly with the size of the last-layer parameters and the validation set, and is largely decoupled from the depth or width of the backbone.
> >
> > To give a clearer sense of magnitude, we report typical training time and resource usage on Waterbirds: as shown in Table 13, the total training time for BFR is 16.33 minutes, which is on the same order as DFR (14.60) and GSR (17.34), and much lower than BAM (57.32), while inference GFLOPs, latency, and memory usage remain nearly identical across all four methods. These findings suggest that the additional overhead introduced by BFR is mild and manageable. For detailed numerical results and further discussion, please refer to the global response and Appendix D.6.
> >
> > ---
> >
> > > **Last  Layer fine-tuning**: The Bayesian Last Layer is efficient ...A discussion or comparison illustrating this trade-off would be valuable.
> >
> > We thank the reviewer for raising this important question. In our framework, the second stage is deliberately designed as a lightweight adaptation step on top of a pretrained backbone, rather than full model retraining. This design choice is consistent with a common paradigm in recent work on group robustness and distribution shift, where a pretrained backbone is frozen and only the last layer is retrained or reweighted (e.g., DFR, GSR), achieving strong worst-group performance with low computational overhead.
> >
> > In our experiments, this localized adaptation does not appear to impose a performance bottleneck. As shown in Table 1 in the main text, BFR **already outperforms methods that update many more parameters or approximate full-parameter retraining** (e.g., JTT, CNC, BAM).
> > To further probe the trade-off between adaptation scope and robustness, we performed an additional ablation where we gradually unfreeze more layers in Stage 2 while keeping the uncertainty-guided bi-level optimization unchanged. On Waterbirds, we denote by “0” the default BFR setting (only the last layer is updated), and by “1–4” the variants that unfreeze the last 1–4 additional layers. The results are:
> >
> > | Waterbirds (BFR) | 0 (last layer) | 1      | 2      | 3      | 4      |
> > |------------------|----------------|--------|--------|--------|--------|
> > | WGA              | 92.43          | 86.60  | 87.23  | 85.83  | 85.83  |
> > | ACC              | 94.63          | 95.51  | 95.10  | 95.18  | 95.18  |
> >
> > We observe that unfreezing additional layers leaves mean accuracy essentially unchanged (or slightly higher), but consistently reduces WGA compared to the last-layer-only BFR. This suggests that, in our uncertainty-guided bi-level scheme and backbone setting, keeping the backbone fixed acts as a useful regularizer that preserves its representation while allowing the Bayesian last layer to perform targeted corrections.
> >
> > From the computational side, the global response (Computation analysis) further illustrates the efficiency aspect of this trade-off. On Waterbirds dataset, **a full-parameter fine-tuning method like BAM is more than 3× slower than BFR on Waterbirds while still underperforming it in WGA**. This indicates that restricting Stage-2 updates to the Bayesian last layer offers a favorable robustness–efficiency trade-off rather than a limiting factor in our framework.
> >
> > We will add the above comparison and discussion to the revised manuscript in **Appendix C.4.2** to make this trade-off explicit.

---

> > > ### Author Response · Authors · 2025-11-19
> > >
> > > > **Q3**:It would be helpful to include or discuss results on how sensitive BFR’s performance is to hyperparameters.
> > >
> > > We sincerely thank the reviewer for the thoughtful suggestions. Overall, our experiments indicate that BFR is reasonably robust to hyperparameter choices and that the largest robustness gains come from (i) using full predictive uncertainty (vs. aleatoric/epistemic alone) and (ii) the bi-level optimization stage.
> > >
> > > Concerning hyperparameter sensitivity, we have already conducted a systematic analysis in **Appendix D.2 and D.4**. On Waterbirds, we performed a $3\times3\times3$ grid search over the inner learning rate, the $p$-value threshold, and random seeds, and compared BFR (with bi-level optimization) against Direct-U, which directly reweights samples by uncertainty. As shown in **Fig. 10**, under the same hyperparameter grid, the distributions of worst-group accuracy and test loss for BFR are considerably more concentrated and shifted toward better regions, whereas Direct-U exhibits much larger variance. This indicates that BFR is substantially less sensitive to hyperparameters and achieves more stable convergence. Similarly, **Fig. 6 in Appendix D.2** shows that varying the $p$-value threshold (0.01 / 0.05 / 0.10) leads to relatively smooth changes in BFR’s worst-group accuracy, with 0.05 achieving the best or near-best performance on most datasets. **We have made these findings more explicit in the main text in line 475-479 and summarize it in the beiginning of Appendix D**.
> > >
> > > Regarding “which components contribute most,” we analyze this question from the perspectives of uncertainty formation and training dynamics. First, we added an ablation on different uncertainty components on Waterbirds, constructing three variants: using only aleatoric uncertainty ($\theta$), using only epistemic uncertainty ($\Phi$), and using the combined predictive uncertainty to drive bi-level optimization. The results are as follows:
> > >
> > >
> > > | Uncertainty        | Waterbirds WGA       | Waterbirds Mean      | CelebA WGA           | CelebA Mean          | MNLI WGA             | MNLI Mean            | Civil WGA            | Civil Mean           |
> > > |----|-------|-----|--------|-----------|--------|------|------------|------|
> > > | ERM                | 71.9 ± 1.5          | 91.4 ± 1.7            | 45.1 ± 0.8            | 95.1 ± 0.4            | 59.2 ± 0.3            | 81.9 ± 0.1            | 54.6 ± 0.6            | 91.4 ± 0.1            |
> > > | Aleatoric ($\theta$)      | 85.98 ± 0.6          | 95.58 ± 0.3           | 78.33 ± 0.4           | 92.26 ± 0.2           | 70.12 ± 0.8           | 80.60 ± 0.2           | 67.91 ± 0.7           | 91.41 ± 0.3           |
> > > | Epistemic ($\Phi$)      | 87.38 ± 0.9          | 95.65 ± 0.3           | 68.89 ± 0.6           | 92.93 ± 0.3           | 70.91 ± 0.9           | 80.55 ± 0.3           | 73.64 ± 0.8           | 90.32 ± 0.2           |
> > > | **Predictive (BFR)** | **92.8 ± 1.5**      | 95.1 ± 0.4            | **83.9 ± 1.0**        | 91.2 ± 0.7            | **73.8 ± 0.9**        | 80.8 ± 0.5            | **80.5 ± 1.6**        | 88.2 ± 0.3            |
> > >
> > >
> > > It shows that the combined uncertainty yields an additional 5–7 percentage points improvement in worst-group accuracy over the single-component variants, while maintaining similar mean accuracy. This suggests that, within our framework, the posterior uncertainty learned in the probabilistic pretraining phase is not only effective by itself, but that the joint use of aleatoric and epistemic components forms one important source of the robustness gains.
> > >
> > > Second, we further assess the contribution of different training stages. Based on **Table 1 (main text) and Appendix Table 11**, we compare the model after probabilistic pretraining alone (ERM-BFR) with standard ERM. ERM-BFR already improves both worst-group accuracy and calibration, indicating that the posterior uncertainty obtained in the first stage contributes positively to robustness. Moreover, keeping the uncertainty estimation fixed, **Fig. 10 in Appendix D.5** compares Direct-U with full BFR under the same hyperparameter grid. The clear advantages of BFR in both performance and stability demonstrate that the bi-level scheme is crucial for transforming uncertainty into a reliable and optimizable training signal.
> > >
> > > We integrated and highlight these findings in the revised version in **Appendix D** to give readers a clearer understanding of BFR’s hyperparameter robustness and the relative contributions of its components.
> > >
> > > ### ***References***:
> > >
> > > [1] Simple and fast group robustness by automatic feature reweighting, ICML 2023.
> > >
> > > [2] Just train twice: Improving group robustness without training group information, ICML 2021.
> > >
> > > [3] Group-robust Sample Reweighting for Subpopulation Shifts via Influence Functions, ICLR 2025.
> > >
> > > [4] Diverse Prototypical Ensembles Improve Robustness to Subpopulation Shift, ICML 2025.
> > >
> > > [5] Bias Amplification Enhances Minority Group Performance, TMLR 2024.

---

### Official Review · Reviewer_8Xr1 · 2025-11-01

**Soundness:** 3
**Presentation:** 3
**Contribution:** 2
**Rating:** 6
**Confidence:** 4

**Summary:**

The paper proposes Bayesian Feature Reweighting (BFR), which uses a Bayesian Last Layer to estimate calibrated uncertainty, partitions validation data into low- and high-uncertainty groups, and then applies bi-level optimization to stably reweight training and improve robustness.

**Strengths:**

1. Combining the Bayesian Layer for Uncertainty and using the uncertainty as a proxy for group label is interesting
2. The experimental results are solid and extensive.

**Weaknesses:**

1. Using a Bayesian last layer for uncertainty estimation and reweighting the loss via (pseudo) group labels is established in prior work; the novelty here appears to lie primarily in the specific combination.

**Questions:**

1. Is the Bayesian last layer strictly necessary? For group partitioning, could alternative uncertainty proxies (e.g., loss, confidence, MC uncertainty) work?
2. What is the computation efficiency of the proposed methods?
3. Figure 2 shows the correlation between uncertainty and group labels in the visualization. Is it possible to calculate the correlation quantitatively?

---

> ### Author Response · Authors · 2025-11-19
>
> We sincerely thank the reviewer for the insightful suggestions and positive assessment of our work, below we address them in details:
>
> **Summary**:
>
> - We further clarify that the main contribution of this work lies in enabling ***predictive uncertainty to serve as a stable training-time control signal in a group-annotation-free setting***.
>
> - We systematically compare the Bayesian last layer with ***loss-, entropy-, and MC-dropout–based uncertainty proxies***.
>
> - We supplement the paper with an analysis of ***GFLOPs, latency, memory usage, and total training time***, demonstrating that BFR has computational overhead broadly comparable to existing two-stage methods.
>
> - We also provide ***a quantitative ROC-AUC measurement*** for the correlation between uncertainty and minority groups in ***Figure 2***, offering more direct evidence that the learned uncertainty carries implicit group information.
>
> ------
>
> > **W1**: Using a Bayesian last layer for uncertainty estimation and reweighting the loss via (pseudo) group labels is established in prior work; the novelty here appears to lie primarily in the specific combination.
>
> Thank you for the thoughtful attention to the aspects of innovation. We agree that Bayesian last layers and group-based reweighting have both appeared in prior work, and our goal is not to claim novelty at the level of individual modules. Instead, the contribution of this paper lies in how these ingredients are organized into a training-time mechanism that works in a  **group-annotation-free** setting, where only the model’s own uncertainty is available to guide learning.
>
> In our formulation, the components are coupled through explicit dependencies rather than being used in isolation. The Bayesian last layer does not only provide predictive uncertainty—it also yields decomposable epistemic and aleatoric information via posterior sampling, allowing us to perform statistical significance tests on the sampled distributions. The resulting information is not used merely as a heuristic metric; rather, it directly determines which samples are treated as relatively trustworthy "support" signals and which as more challenging "target" signals, thereby explicitly shaping the outer-level weight adjustment in the bi-level optimization.
>
> Furthermore, the sparsity induced by the Gamma prior in the Bayesian last layer naturally concentrates the inner-loop updates on a small subset of parameters that are most sensitive to uncertainty. This prior-induced sparsity restricts the influence of weight adjustments and avoids the gradient instabilities that arise when uncertainty is used for direct weighting. Thus, the localized updates are not an artificial constraint but an intrinsic consequence of the probabilistic modeling choice, allowing the significance-based sample partition to propagate through the bi-level optimization in a stable and controlled way. These design choices create a tight functional coupling among posterior sampling, significance testing, and bi-level optimization, enabling uncertainty to shape training in a statistically grounded and interpretable manner, rather than through heuristic reweighting.
>
> From this perspective, our contribution goes beyond a specific combination of existing elements. We introduce a training mechanism that operates genuinely without group annotations and allows predictive uncertainty to serve as a reliable training-time control signal. The design encompasses how uncertainty is generated (Sec. 3.1), how it is transformed into statistically meaningful decision cues (Sec. 3.2), and how it interacts with optimization in a numerically stable manner (Sec. 3.4/3.5). In particular, we conduct a systematic analysis of the gradient instabilities caused by naive uncertainty-based weighting (Appendix D.4), which motivates the structural choices in our framework. Consequently, the proposed method is not merely a recombination of modules, but a reproducible, editable, and stable training paradigm tailored to the group-annotation-free setting.
>
> Once again, we sincerely thank the reviewer for the careful reading and valuable feedback. Your comments have helped us clarify the motivation and structure of our method and guided us to further elaborate on the coupling among its components. We truly appreciate your time and consideration.

---

> > ### Author Response · Authors · 2025-11-19
> >
> > > **Q1**: Is the Bayesian last layer strictly necessary? For group partitioning, could alternative uncertainty proxies (e.g., loss, confidence, MC uncertainty) work?
> >
> > Regarding this concern, we will clarify the core rationale behind choosing BLL in our framework, followed by additional experiments that directly support our claims.
> >
> > First, the purpose of adopting BLL in our framework goes beyond obtaining stable uncertainty estimates. BLL plays a dual role: (1) providing well-calibrated predictive uncertainty, and (2) inducing parameter-level sparsity through controllable priors, which allows the model to update only a very small, targeted subset of weights during the second-stage fine-tuning. This ensures editability while preventing unintended disruption to previously learned representations. Such behavior is difficult to obtain with loss-based or entropy-based proxies, which do not induce a structured posterior over parameters; and although logit-variance (MC-dropout) uncertainty may create sparse activations during forward passes, it does not yield a controllable, structured posterior over parameters. As a result, it cannot constrain the gradient flow during fine-tuning, making it difficult to realize similarly localized and minimal parameter updates.
> >
> > To further address the your question, we conducted two sets of experiments to systematically compare BLL against different classes of uncertainty proxies:
> >
> > (1) Comparison of calibration quality and robustness gains.
> > We include three representative baselines: (i) a loss-based hard-sample–selection method [1]—also included in Table 1 of the main paper—which identifies bias-conflicting samples using a bias-only model and therefore represents a canonical loss-driven uncertainty proxy; (ii) a confidence-based proxy using the entropy of ERM outputs; and (iii) an dropout-based uncertainty proxy using dropout sampling. To ensure fairness, all proxies use the same ERM backbone, and we only replace the uncertainty estimation method. We then evaluate worst-group accuracy (WGA) and mean accuracy on the Waterbirds and CelebA datasets, following the same experimental protocol as Table 1. The results are as follows:
> >
> >
> >
> > | Uncertainty Proxy    | Waterbirds WGA      | Waterbirds Mean ACC | CelebA WGA         | CelebA Mean ACC     |
> > |----|-----|----|---|----|
> > | LfF (loss-based)    | 78.0   | 91.2                  | 77.2                | 85.1                  |
> > | Entropy-based        | 88.2 ± 1.1           | 96.2 ± 0.5            | 76.1 ± 0.9          | 92.0 ± 0.4            |
> > | Dropout-based        | 87.2 ± 0.8           | 95.9 ± 0.5            | 68.3 ± 1.2          | 93.0 ± 0.6            |
> > | **BFR (Ours)**       | **92.8 ± 1.5**       | **95.1 ± 0.4**        | **83.9 ± 1.0**      | **91.2 ± 0.7**        |
> >
> > As shown in the table, uncertainty proxies based on loss, entropy, or dropout consistently achieve weaker worst-group performance than BFR, while their mean accuracy remains at a similar level. This indicates that although these alternative proxies can preserve overall accuracy, they are less effective at identifying bias-conflicting samples and improving worst-group accuracy. In contrast, BFR with a Bayesian last layer yields more stable and consistently stronger robustness gains across both datasets, suggesting that the uncertainty signal provided by BLL is more effective for subgroup partitioning and sample selection in our framework.
> >
> > (2) Comparison of parameter editability.
> > We further analyze the extent of parameter changes before and after uncertainty-guided bi-level optimization when using confidence-based and MC-based uncertainty proxies. Following the metric used in Table 3 of the main paper, we sample one run per model from multiple random runs of each Stage-1 proxy model, and compute the proportion of last-layer weights whose absolute change exceeds a given threshold on waterbirds datasets(lower values indicate more localized parameter updates). The results are as follows:
> >
> > | waterbirds     | 0       | 1e-4    | 1e-3    | 1e-2    |
> > |----|----|----|----|----|
> > | Entropy-based  | 100     | 99.82   | 96.24   | 64.54   |
> > | Dropout-based  | 100     | 99.85   | 97.68   | 75.81   |
> > | **BFR (Ours)** | **58.39** | **58.22** | **56.39** | **28.70** |
> >
> >
> > The results show that entropy- and dropout-based proxies lead to substantially larger and more widespread parameter updates. In contrast, BFR due to its sparsity-inducing priors and structured posterior which modifies only a small, well-targeted subset of parameters during retraining, thus markedly improving editability and reducing collateral interference.
> >
> > In summary, within our framework BLL is not merely an uncertainty estimator; rather, it is a mechanism that simultaneously enables (i) stable, instance-level uncertainty guidance and (ii) sparse, localized, and controllable parameter editing. We will incorporate the above comparative experiments and further analysis into the **Appendix D.2 (Table 12 and 13)**.

---

> > > ### Author Response · Authors · 2025-11-19
> > >
> > > > **Q2**:  What is the computation efficiency of the proposed methods?
> > >
> > > Thank you for raising this specific question regarding computational efficiency. As summarized in our **global response (Computation Analysis)**, we have provided both theoretical and empirical clarifications on the computational cost of BFR. Theoretically, Sec. 3.4 analyzes that BFR’s additional overhead relative to standard two-stage frameworks arises solely from performing a limited number of updates on the Bayesian last layer during the second stage. Empirically, we conducted unified measurements of GFLOPs, inference latency, inference-time memory consumption, and total training time for DFR, BAM, GSR, and BFR on the Waterbirds dataset (Appendix D.6, Table 18). The results show that BFR’s inference cost is nearly identical to existing methods, and its total training time is comparable to DFR and GSR while being substantially lower than BAM. These findings indicate that in practice, BFR provides robustness improvements while maintaining computational efficiency on par with other two-stage approaches. For additional details and numerical comparisons, please refer to the global response (computation analysis).
> > >
> > > ---
> > >
> > > > **Q3**: Figure 2 shows the correlation between uncertainty and group labels in the visualization. Is it possible to calculate the correlation quantitatively?
> > >
> > > To further quantify the correlation between predictive uncertainty and latent group structure shown in Figure 2, we evaluate the ROC-AUC on the test set with available group labels by treating minority groups as the positive class and majority groups as the negative class, using the instance-level predictive uncertainty as the scoring function. In this binary discrimination setting, an AUC greater than 0.5 indicates performance above random chance and therefore reflects non-trivial information content [2]. On the Waterbirds dataset, the AUC is **0.87**, indicating that the uncertainty score more frequently assigns higher values to minority-group samples than to majority-group ones. In other words, the uncertainty estimate carries meaningful discriminative information about the underlying subgroups rather than behaving like random noise. We have incorporated this quantitative result into the **description of Figure 2 in Sec 4.1**. Thank you for the helpful suggestion.
> > >
> > > ### ***References***:
> > >
> > > [1] Learning from failure: De-biasing classifier from biased classifier. NeurIPS 2020
> > >
> > > [2] A Baseline for Detecting Misclassified and Out-of-Distribution Examples in Neural Networks. ICLR 2017

---

### Official Review · Reviewer_3fkM · 2025-11-03

**Soundness:** 3
**Presentation:** 4
**Contribution:** 1
**Rating:** 2
**Confidence:** 5

**Summary:**

The paper introduces an uncertainty-driven training framework that leverages probabilistic pretraining and statistical test-based uncertainty estimation. Rather than relying on a fixed uncertainty-weighting scheme, the approach uses the estimated uncertainties to partition the training data into support and target sets. A bi-level optimization strategy is then employed, where the inner loop estimates sample weights and the outer loop updates the last-layer parameters. The proposed method is evaluated across diverse datasets spanning image, language, and distribution-shift scenarios.

**Strengths:**

**1. Well-motivated:** The paper is well-motivated and presents a well-crafted motivating discussion on both classical and SOTA methods of uncertainty estimation and spurious correlation.

**2. Well-presented:** The paper is very well-organized in terms of describing its technical parts such as probabilistic pretraining, uncertainty estimation, and the proposed bi-level optimization scheme.

**3. Diverse experimental settings:** The paper employs diverse datasets including image, language, and distribution shift to validate their proposed method.

**Weaknesses:**

**1. Absence of Innovation in Sect 3.2:** This section essentially reiterates the theory proposed by [1]. The derivation of Eqn. 6 in Appendix C is already presented in detail in Eqn (13) of [1]. Therefore, Eqn. (6) can be directly cited from [1] without the claimed novelty in the writing.

**2. Limited novelty in Sect 3.3.** This section is essentially a derivative of the original p-valued based uncertainty estimation of [2]. The bi-level optimization was also originally proposed by [3] which also adopts similar bi-level training strategy to weight the samples based on uncertainty.

[1] Xinyue Hu, Zhibin Duan, Bo Chen, and Mingyuan Zhou. Enhancing uncertainty estimation and interpretability with bayesian non-negative decision layer. In The Thirteenth International Conference on Learning Representations, 2025.

[2] Xinjie Fan, Shujian Zhang, Korawat Tanwisuth, Xiaoning Qian, and Mingyuan Zhou. Contextual dropout: An efficient sample-dependent dropout module. In International Conference on Learning Representations, 2021

[3] Xiao Zhou, Yong Lin, Renjie Pi, Weizhong Zhang, Renzhe Xu, Peng Cui, and Tong Zhang. Model agnostic sample reweighting for out-of-distribution learning. In International conference on machine learning, pp. 27203–27221. PMLR, 2022.

**Questions:**

The paper is clearly written and well-presented, with comprehensive experimental results. However, its main contribution appears to be an effective engineering integration of the theories proposed in [1], [2], and [3], rather than a fundamentally novel methodological advancement. I recommend that the authors strengthen the technical and theoretical aspects of the work to make it more suitable for theory-oriented venues such as ICLR, ICML, or NeurIPS.

---

> ### Author Response · Authors · 2025-11-19
>
> > **W1**:  Absence of Innovation in Sect 3.2: This section essentially reiterates the theory proposed by [1]. The derivation of Eqn. 6 in Appendix C is already presented in detail in Eqn (13) of [1]. Therefore, Eqn. (6) can be directly cited from [1] without the claimed novelty in the writing.
>
> We sincerely thank the reviewer for the constructive feedback on Section 3.2. Regarding the comment that “Section 3.2 lacks innovation and Eq. (6) can be directly cited from [1],” we would like to respectfully clarify that we do not agree with this conclusion. Eq. (6) follows the standard notation and decomposition of the Evidence Lower Bound (ELBO), whose mathematical form and symbolic conventions are widely adopted across variational inference and generative modeling literature (e.g., [4, 5]). The purpose of presenting Eq. (6) in our Section 3.2 is **not to claim novelty in the ELBO itself**, but to produce a **training-time uncertainty control signal** that supports the annotation-free bi-level robust optimization in Sec. 3.3 and the outer-gradient differentiability analysis in Sec. 3.5. This design is fundamentally different from [1], where the Bayesian layer is employed for inference-time interpretability and calibration.
>
> Therefore, the inclusion of Eq. (6) and the exposition in Section 3.2 are intended to ensure internal consistency between the probabilistic head and the subsequent training framework. Under our notation and prior assumptions, this instantiation is a necessary bridge that aligns the Bayesian head with the outer bi-level objectives (Eqs. 10–11). Directly citing [1] without such instantiation would obscure the logic connecting posterior–prior matching and the uncertainty-derived control signal.
>
> Following the reviewer’s helpful suggestion, and to avoid any potential misunderstanding, we will make the following revisions in the updated manuscript:
>
> 1. Condense Appendix C, replacing redundant derivations with concise references, while retaining only the components directly relevant to BFR’s training-time control and bi-level optimization.
>
> 2. Clarify in the contribution statement that the innovation of this work lies in integrating Bayesian uncertainty as a controllable signal into the group annotation-free bi-level robust optimization, rather than in the ELBO formulation or its derivation itself.

---

> ### Author Response · Authors · 2025-11-19
>
> >**W2**: Limited novelty in Sect 3.3. This section is essentially a derivative of the original p-valued based uncertainty estimation of [2]. The bi-level optimization was also originally proposed by [3] which also adopts similar bi-level training strategy to weight the samples based on uncertainty.
>
> We thank the reviewer for the comments on Section 3.3. We would like to clarify that the proposed “Uncertainty-guided Bi-level Optimization’’ is not a direct combination or straightforward extension of [2] or [3]. Instead, it is a framework designed specifically around a central question: how can one achieve group-robust training in a fully group-annotation-free setting using only the model’s own predictive uncertainty? Our methodological contributions can be understood on three levels.
>
> First, in terms of objective and application setting, our framework is among the few that can improve group robustness without any form of group supervision, relying solely on instance-level predictive uncertainty produced by the model itself. Prior work such as [3] requires explicit group labels—or at least validation-time group annotations—whereas our method does not rely on any external labels or pseudo-groups generated by other additional models. Instead, the support and target sets are constructed directly from the posterior-based uncertainty obtained during Probabilistic Pretraining, which then drives the subsequent bi-level optimization. Existing literature provides no systematic training procedure for enabling uncertainty to play an operational role in group-robust learning under a group-free regime, and our work fills this gap by offering a reproducible training-time pipeline.
>
> Second, at the level of methodological design, our approach diverges from both [2] and [3] in essential ways:
>
> 1. Regarding [2], the $p$-value in Contextual Dropout is used to measure differences between dropout activations, serving the purpose of enhancing sample-dependent dropout behavior. In our work, the $p$-value is used solely as a statistical testing tool to assess whether the top-2 posterior predictive probabilities differ significantly, which in turn provides a principled control signal for forming the support and target sets. Moreover, unlike [2], our uncertainty arises from posterior sampling in a probabilistic last layer, with a clear aleatoric and epistemic decomposition and well-defined probabilistic semantics. This gives the resulting partition and the bi-level updates a more explicit statistical grounding. **Our framework does not incorporate contextual dropout, nor do we use $p$-values as part of network structure or regularization; therefore, the purpose and role of $p$-value in our method are different in objective and usage from those in [2].**
>
> 2. Regarding [3], our inner–outer objectives are not based on MAPLE-style approximate backpropagation-through-trajectory. Instead, we employ differentiable implicit gradients, allowing the influence of sample weights to propagate only through the inner-level optimum, which ensures numerical stability in the absence of group supervision. Furthermore, unlike [3], our inner loop optimizes only the Bayesian Last Layer and does so under sparsity, enabling localized and editable updates that substantially limit parameter perturbation (as shown in Table 3). **We additionally report a direct comparison of computational cost, showing that our approach is significantly more efficient than [3]**.
>
> Third, from the perspective of optimization difficulty, using uncertainty directly to guide robust training is far from trivial. As detailed in the main text and Appendix D.4, naïvely weighting samples by uncertainty leads to severe convergence issues—high-uncertainty samples cause gradient explosion, whereas low-uncertainty samples yield nearly vanishing gradients. Our bi-level formulation explicitly resolves this instability by dynamically adapting the sample weights, which is empirically validated through grid-search sensitivity analyses (Appendix D.4, Fig. 9). Thus, our method is not another instance of “uncertainty-based reweighting,’’ but a structural solution to its fundamental optimization limitations.
>
> In summary, the proposed method is not a reformulation of [2] or [3]. Rather, it establishes a new uncertainty-driven training pathway that remains stable and statistically grounded without requiring any group annotations. We have revised Section 3.3 and Appendix C.4 to further clarify these distinctions and to highlight the independent methodological contributions of our approach in terms of objective, design, and optimization behavior.

---

> > ### Author Response · Authors · 2025-11-19
> >
> > ### ***References***:
> >
> > 1. Xinyue Hu, Zhibin Duan, Bo Chen, and Mingyuan Zhou. *Enhancing uncertainty estimation and interpretability with Bayesian non-negative decision layer.* In *The Thirteenth International Conference on Learning Representations (ICLR)*, 2025.
> >
> > 2. Xinjie Fan, Shujian Zhang, Korawat Tanwisuth, Xiaoning Qian, and Mingyuan Zhou. *Contextual dropout: An efficient sample-dependent dropout module.* In *International Conference on Learning Representations (ICLR)*, 2021.
> >
> > 3. Xiao Zhou, Yong Lin, Renjie Pi, Weizhong Zhang, Renzhe Xu, Peng Cui, and Tong Zhang. *Model agnostic sample reweighting for out-of-distribution learning.* In *International Conference on Machine Learning (ICML)*, pp. 27203–27221. PMLR, 2022.
> >
> > 4. Taejong Joo, Uijung Chung, and Min-Gwan Seo. *Being Bayesian about categorical probability.* In *International Conference on Machine Learning (ICML).* PMLR, 2020.
> >
> > 5. Diederik P. Kingma and Max Welling. *Auto-encoding variational Bayes.* In *International Conference on Learning Representations (ICLR)*, 2014.

---

### Official Review · Reviewer_g921 · 2025-11-04

**Soundness:** 2
**Presentation:** 2
**Contribution:** 2
**Rating:** 4
**Confidence:** 4

**Summary:**

The paper proposes Bayesian Feature Reweighting (BFR), which replaces the classifier with a Bayesian Last Layer (BLL) to get calibrated, instance-level uncertainty under sparsity-promoting priors, then uses that uncertainty to control training via a bi-level optimization that upweights high-uncertainty examples while keeping parameter edits localized to the head. Evaluated on Waterbirds, CelebA, MultiNLI, CivilComments, and an ImageNet-9 to ImageNet-A shift. BFR improves worst-group accuracy and shows smaller fractions of changed parameters vs last-layer retraining baselines.

**Strengths:**

Experiments demonstrate some improvement in worst-group and OOD accuracy across vision (Waterbirds, CelebA, ImageNet-A) and language (MultiNLI, CivilComments) tasks.

**Weaknesses:**

- The novelty of this paper is incremental: BLLs and bi-level reweighting are established methods. The contribution is mainly coupling a known BLL with a bi-level sample-reweighting loop. There is little new probabilistic machinery beyond the proposed method.
- The paper claims to "turn uncertainty into control", but the control mechanism is largely heuristic partitioning of data into "support" vs "target" via that p-value, followed by a standard min–max bi-level update. There is no derivation that this procedure optimizes worst-group risk under the proposed uncertainty.
- The introduction argues that principled ways to use uncertainty as a training signal to enhance representation learning remain underexplored, yet the method offered does not provide calibration guarantees or a decision-theoretic objective tying the p-value to group-robust risk, so the “principled” claim is not met.
- The paper asserts aleatoric and epistemic uncertainty modeling, but validates only a unified predictive uncertainty.
- Computational cost (e.g., time, FLOPs, or memory usage) is not reported to support the "lightweight" claim.

**Questions:**

See weaknesses.

---

> ### Author Response · Authors · 2025-11-19
>
> We thank the reviewer for the thoughtful comments and careful assessment of our work.  Our detailed responses are as follows:
>
> **Summary:**
>
> - We have clarified the scope and novelty by positioning our contribution as a structured way to use Bayesian last-layer uncertainty as a ***training-time control signal in a group-annotation-free*** setting.
> - We have added ***ablations on aleatoric, epistemic, and predictive uncertainty*** across four datasets, showing complementary effects and consistent worst-group gains over ERM.
> - We have provided a systematic ***computation analysis (GFLOPs, latency, memory, training time)*** and softened the wording from “lightweight” to “comparable computational overhead”.
> ----
> >**W1**: The novelty of this paper is incremental: BLLs and bi-level reweighting are established methods. The contribution is mainly coupling a known BLL with a bi-level sample-reweighting loop. There is little new probabilistic machinery beyond the proposed method.
>
> We appreciate this concern about novelty. Our goal is indeed not to introduce a new probabilistic model family, but to address a specific question that has been largely underexplored: **how to turn existing Bayesian last-layer (BLL) machinery into a stable, training-time control signal for group-robust learning when no group annotations are available.** In this sense, the contribution is about a new way of using established probabilistic tools in a challenging setting, rather than proposing new probabilistic primitives.
> More concretely, our perspective consists of three parts:
>
> (1) **From inference-time BLL to training-time control**. First, Prior work typically uses BLLs as an inference-time tools rather than training-time control signals. In most BLL-based uncertainty estimation or calibration work, the posterior is used at test time for epistemic/aleatoric decomposition [1], constructing predictive confidence, or posterior averaging [2, 3], but it is not used to influence the training dynamics themselves. In contrast, our setting requires posterior uncertainty to be transformed into a differentiable training-time decision signal—a usage that is rarely explored in prior BLL literature.
>
> (2) **Improving group robustness without any group or pseudo-group labels**. Most  reweighting approaches rely on explicit or pseudo group information. Representative methods assume access to group labels [4, 5], or at least a group-aware validation signal [6, 7], which is then used to define the outer-level objective. In our fully group-agnostic setting, no such supervision—neither explicit nor pseudo-group labels—is allowed. This renders conventional bi-level approaches inapplicable and necessitates a new way to construct sample-dependent training signals, including how to form meaningful “support/target” splits without any group information.
>
> (3) **Structured use of uncertainty instead of direct uncertainty weighting.** Directly using uncertainty as sample weights leads to unstable learning [8]. Instead of scaling the loss by uncertainty magnitude, we combine the significance-based sample partition with the sparsity-inducing prior in the Bayesian last layer, which causes inner-loop updates to concentrate on a small subset of parameters most relevant to uncertainty. The resulting updates can then be stably propagated to the outer objective through differentiable bi-level optimization. The experiments in Appendix D.5 show that, compared with direct uncertainty weighting, our bi-level formulation yields more consistent and stable convergence, supporting the training benefits of this structured design.
>
> From the perspective of probabilistic machinery, we do not claim to introduce a novel probabilistic model family, nor is this the intention of our work. Our focus is on enabling established probabilistic tools to function effectively in a setting where they traditionally do not: by using carefully designed posterior-based significance testing, a sparsity-inducing prior, and interfaces to bi-level optimization, we allow these existing probabilistic components to **operate coherently as training-time control signals in a fully group-agnostic group-robust learning task**. This creates a statistically grounded and functionally integrated pathway—from posterior uncertainty to sample partition to outer-level optimization—that goes beyond a simple combination of modules. We believe this structural use of existing probabilistic machinery in a new training regime constitutes a meaningful contribution.

---

> > ### Author Response · Authors · 2025-11-19
> >
> > > **W2**: The paper claims to "turn uncertainty into control", ... under the proposed uncertainty.
> >
> > Regarding the concerns about the phrase “turning uncertainty into control.” Our intention is not to claim formal equivalence to an oracle optimizer of the worst-group risk; rather, we emphasize that predictive uncertainty is explicitly incorporated during training and used as a control signal. Concretely, the $p$-value is not a purely heuristic splitter but the outcome of a statistical test on differences between predictive distributions, which quantifies instance-level confidence. The resulting statistic enables a partition that we interpret as an explicit uncertainty-conditioned decomposition of the data distribution (Eqs. 8–11):
> >
> > $$ p(X) = p(X \mid U(x) < \delta) + p(X \mid U(x) \ge\delta),$$
> >
> > where $U(x)$ denotes the uncertainty estimator and $\delta$ is the corresponding threshold. In other words, the original distribution $p(X)$ is decomposed into a high-confidence component and a low-confidence component, aligning the support vs. target split with distinct posterior regions for low- vs. high-confidence samples.
> >
> > Building on this decomposition, the **outer-level min–max objective (Eqs. 10–11) is consistent with the empirical Distributionally Robust Risk Minimization (DRRM) framework [5]**:
> >
> > $$
> > \min_{\omega}  \max_{P \in U(P_0)} E_{(x,y)\sim P} [ l(f_\omega(x), y) ]
> > $$
> >
> >
> > where $U(P_0)$  is an uncertainty set centered at the empirical distribution $P_0$. The key difference is that classical DRRM instantiations typically rely on explicit group annotations to construct $U(P_0)$, whereas our method induces implicit subgroups from model uncertainty (Eqs. 10–11), achieving worst-group robustness without group labels.
> > Moreover, in Section 3.5 (Theoretical Analysis) we show that, under standard regularity assumptions (smoothness, invertible Hessian, and differentiability required for the implicit function theorem), the dependence of the outer-level gradient on the inner optimum can be formalized via implicit differentiation, yielding updates that follow the descent direction for the worst-case risk under the target distribution. Hence, while the support/target construction is grounded in a statistical test, the overall training procedure remains a theoretically differentiable and stable bi-level optimization scheme.
> >
> > We will clarify these points in the revision by: detailing the link between the $p$-value–based partition and an upper bound on worst-group risk in Sec 3.3, and making precise that “turning uncertainty into control” refers to using calibrated uncertainty as a training-time control variable to modulate updates in a stable, localized manner—rather than a mere heuristic split.
> >
> > ---
> >
> > > **W3**: The introduction argues that principled ways ..., so the “principled” claim is not met.
> >
> > As for the use of the term “principled uncertainty usage.” We appreciate that, in a strict sense, “principled” is often reserved for frameworks endowed with formal calibration guarantees or explicit decision-theoretic objectives. While we agree with this interpretation, that direction lies beyond the scope of the present work. In our paper, “principled” is intended to denote a systematic and interpretable design rather than a claim of theoretical optimality. Our goal is not to present a fully decision-theoretic model with formal guarantees, but to investigate a systematic, statistically interpretable, and practically viable way to leverage uncertainty during training.
> >
> > Specifically, our method formally quantifies instance-level uncertainty and uses it as a control signal within a bi-level optimization scheme, thereby converting uncertainty into a controllable training-time constraint. Unlike approaches that rely on heuristic weights or pre-specified losses, our mechanism follows principles of statistical inference and thus provides a structured—hence “principled” in our sense—realization at the modeling level. Empirically, Appendix Table 8 shows that ERM-BFR substantially reduces Expected Calibration Error (ECE) under the same backbone, indicating improved calibration of the resulting signals; moreover, Figure 2 demonstrates that ERM-BFR can identify minority regions without group annotations, suggesting that the model indeed exploits uncertainty to enhance group robustness.
> >
> > We will clarify this scope explicitly in the revised introduction: in this paper, “principled” refers to a systematic framework grounded in statistical uncertainty quantification, rather than a decision-theoretic model with optimal calibration guarantees; this perspective is complementary to, and different in scope from, the stricter notion highlighted by the reviewer.

---

> > > ### Author Response · Authors · 2025-11-19
> > >
> > > > **W4**: The paper asserts aleatoric and epistemic uncertainty modeling, but validates only a unified predictive uncertainty.
> > >
> > > To address this concern, we have conducted additional ablation studies to examine their independent contributions. Specifically, we separately employ the aleatoric uncertainty (captured by the local latent variable $\theta$) and the epistemic uncertainty (captured by the global latent variable $\Phi$) as the control signal in the bi-level optimization, while keeping all other settings unchanged. Using the same datasets and evaluation protocol as in the main text, we compare the Worst-Group Accuracy (WGA) and Mean Accuracy (Mean) under each variant. The results are summarized in the following table:
> > >
> > >
> > > | Uncertainty        | Waterbirds WGA       | Waterbirds Mean      | CelebA WGA           | CelebA Mean          | MNLI WGA             | MNLI Mean            | Civil WGA            | Civil Mean           |
> > > |--------------------|----------------------|------------------------|------------------------|------------------------|------------------------|------------------------|------------------------|------------------------|
> > > | ERM                | 71.9 ± 1.5          | 91.4 ± 1.7            | 45.1 ± 0.8            | 95.1 ± 0.4            | 59.2 ± 0.3            | 81.9 ± 0.1            | 54.6 ± 0.6            | 91.4 ± 0.1            |
> > > | Aleatoric ($\theta$)      | 85.98 ± 0.6          | 95.58 ± 0.3           | 78.33 ± 0.4           | 92.26 ± 0.2           | 70.12 ± 0.8           | 80.60 ± 0.2           | 67.91 ± 0.7           | 91.41 ± 0.3           |
> > > | Epistemic ($\Phi$)      | 87.38 ± 0.9          | 95.65 ± 0.3           | 68.89 ± 0.6           | 92.93 ± 0.3           | 70.91 ± 0.9           | 80.55 ± 0.3           | 73.64 ± 0.8           | 90.32 ± 0.2           |
> > > | **Predictive (BFR)** | **92.8 ± 1.5**      | 95.1 ± 0.4            | **83.9 ± 1.0**        | 91.2 ± 0.7            | **73.8 ± 0.9**        | 80.8 ± 0.5            | **80.5 ± 1.6**        | 88.2 ± 0.3            |
> > >
> > > From the table, we observe that across multiple benchmarks, using either aleatoric or epistemic uncertainty alone already yields non-trivial robustness gains, and their effects are complementary: combining them into the full predictive uncertainty consistently achieves the best worst-group accuracy, with an average improvement of about 5–7 percentage points. This indicates that, in our framework, the use of predictive uncertainty is not a simple additive combination of aleatoric and epistemic terms, but rather a natural exploitation of their joint posterior, which is more effective for identifying and emphasizing high-risk samples during training. In addition, we distinguish aleatoric and epistemic uncertainty in the modelling part because our posterior modeling is explicitly built on these two types of uncertainty, while the predictive uncertainty used during training is precisely their unified form.
> > >
> > > We will incorporate this new ablation study and a brief discussion of its implications into **Appendix D.1** of the revised manuscript, in order to more systematically clarify the role of aleatoric and epistemic modeling and their concrete contributions to robustness improvements. We again thank the reviewer for this important suggestion, which has helped us better articulate the design motivation of the posterior structure in BFR and support it with additional empirical evidence.
> > >
> > > ---
> > > > **W5:** Computational cost (e.g., time, FLOPs, or memory usage)
> > >
> > > Thank you for pointing out the lack of quantitative support for the term “lightweight” in the original manuscript. As detailed in our **Global Response (Computation Analysis)**, we have added a systematic evaluation of FLOPs, runtime, and memory overhead in the revised version (Sec. 3.4 and Table 13 in Appendix D.6). Specifically, we compare the GFLOPs, P50/P90 inference latency, inference-time memory usage, and total training time of DFR, BAM, GSR, and BFR under identical hardware and training configurations. The results show that BFR matches these strong baselines almost exactly in inference GFLOPs, latency, and memory usage; its total training time is also on the same order as DFR and GSR, and substantially lower than BAM. Accordingly, we have revised the wording to “comparable computational overhead to existing methods” and now support this claim with the concrete measurements reported in Table 13. For further discussion, please refer to the global response (Computation Analysis).

---

> > > > ### Author Response · Authors · 2025-11-19
> > > >
> > > > ### **References**:
> > > >
> > > > [1] *What Uncertainties Do We Need in Bayesian Deep Learning for Computer Vision?* NeurIPS 2017.
> > > >
> > > > [2] *Enhancing Uncertainty Estimation and Interpretability with Bayesian Non-negative Decision Layer.* ICLR 2025.
> > > >
> > > > [3] *Laplace Redux—Efficient Post-hoc Uncertainty for Deep Networks.* NeurIPS 2021.
> > > >
> > > > [4] *Mode Agnostic Sample Reweighting for Out-of-distribution Learning.* ICML 2022.
> > > >
> > > > [5] *Distributionally Robust Neural Networks for Group Shifts: On the Importance of Regularization.* ICML 2020.
> > > >
> > > > [6] *Last Layer Re-Training is Sufficient for Robustness to Spurious Correlations.* ICLR 2023.
> > > >
> > > > [7] *Meta-Weight-Net: Learning an Explicit Mapping for Sample Weighting.* NeurIPS 2019.
> > > >
> > > > [8] *On the Pitfalls of Heteroscedastic Uncertainty Estimation with Probabilistic Neural Networks.* ICLR 2022.

---

### Author Response · Authors · 2025-11-19
**Global Response**

## Summary of Changes

We thank the area chair and reviewers for their careful reading and helpful feedback. In response, we have made the following main changes and additions to the paper:

1. **Computational cost and scalability.** We added a new section on practical computational analysis in **Appendix D.6 (Table 18)** and referenced it in the main text (Sec. 3.4), reporting wall-clock training time, GFLOPs, inference latency, and memory usage for DFR, BAM, GSR, and BFR on Waterbirds under the same hardware and training setup. These results show that BFR achieves comparable computational cost to other two-stage methods while providing stronger worst-group accuracy.

2. **Ablation on the sparsity prior.**
   We introduced an ablation comparing the non-negative, sparsity-inducing Gamma prior in the Bayesian last layer with a standard Gaussian prior on Waterbirds **(Appendix D.2, Table 14 and 15)**.

3. **Alternative uncertainty proxies and parameter editability.**
   We added experiments that replace the Bayesian last layer with alternative uncertainty proxies, including loss-based, entropy-based, and MC Dropout-based proxies **(Appendix D.2, Table 12 and 13)**.

4. **Ablation on uncertainty types.**
   To clarify the role of probabilistic pretraining and different uncertainty types, we further compare using only aleatoric uncertainty, only epistemic uncertainty, and the full predictive uncertainty from BFR, against a deterministic ERM baseline, on four datasets **(Appendix D.1, Table 9 and 10)**. Both aleatoric and epistemic components improve worst-group accuracy over ERM, while the full predictive uncertainty consistently achieves the best robustness.

5. **Last-layer versus deeper fine-tuning.**
   We conducted an additional ablation that gradually unfreezes more layers in Stage-2 on top of BFR **(Appendix C.4.2, Table 8)**.

6. **Presentation improvements.**
   We revised several parts of the paper to clarify the technical novelty (Sec. 3.2–3.5), improved the discussion of spurious correlation benchmarks (Sec. 4.1), and added additional figures in the appendix to better illustrate the training pipeline and uncertainty signals.

---

## Computation Analysis

We sincerely thank all reviewers for their thorough reading of our work and for the constructive comments. In response to the concerns regarding computational complexity and scalability, we have added an empirical analysis of computational cost in the revised manuscript (Sec. 3.4 of the main text and Table 18 in Appendix D.6). Specifically, on the Waterbirds dataset, we conduct a systematic comparison of four representative group-robust methods—DFR [1], BAM [2], GSR [3], and our proposed BFR—under identical hardware and training configurations. The comparison covers per-image GFLOPs, inference latency (P50/P90), inference-time memory usage, and total wall-clock training time. The results are summarized as follows:

| Waterbirds | GFLOPs | P50 (ms) | P90 (ms) | Memory (MiB) | Total training time (min) |
|-----|--------|-----|----------|---|----|
| DFR       | 0.047  | **4.03** | 4.55     | **232.57**   | **14.60**   |
| BAM       | 0.047  | 4.70     | 4.78     | 232.66       | 57.32|
| GSR       | 0.047  | 4.82     | 5.59     | **232.57**   | 17.34|
| BFR (Ours)       | 0.047  | 4.04     | **4.24** | 232.58       | 16.33 |

1. **Inference efficiency.** As shown above, BFR exhibits virtually identical GFLOPs, P50/P90 inference latency, and memory consumption to DFR, GSR, and BAM. This indicates that the uncertainty-guided bi-level updates only introduce comparable additional inference overhead.

2. **Training cost.** In terms of total training time, BFR (16.33 minutes) is on the same order as GSR (17.34 minutes), slightly higher than DFR (14.60 minutes), and substantially lower than BAM (57.32 minutes). This shows that, even with probabilistic pretraining and bi-level optimization, the practical wall-clock training cost of BFR remains comparable to existing two-stage methods.

These empirical observations are fully aligned with the theoretical analysis in Sec. 3.4: BFR performs updates only on the Bayesian last layer in the second stage, and the additional overhead stems from a small number of gradient steps that scale linearly with the size of the validation set. In practice, this corresponds precisely to the modest training-time differences reported in Table 18.

Overall, we believe these new results demonstrate that BFR achieves robustness gains while maintaining both inference-time and training-time costs within a reasonable and method-comparable range. We hope this additional analysis helps alleviate reviewers’ concerns regarding computational overhead and scalability.

[1] *Last layer re-training is sufficient for robustness to spurious correlations.* ICLR 2023
[2] *Bias amplification enhances minority group performance.* TMLR 2024
[3] *Group-robust Sample Reweighting for Subpopulation Shifts via Influence Functions.* ICLR 2025

---

### Note · Authors · 2026-04-11

I have read and agree with the venue's withdrawal policy on behalf of myself and my co-authors.

---

### Meta-Review · Area_Chair_4qdD · 2025-12-24

**Summary:**

The paper proposes Bayesian Feature Reweighting (BFR), a framework that replaces the classifier at the end of a deep neural network with a Bayesian Last Layer that provides instance-level calibrated uncertainty estimates and imposes sparsity on the decision layer to limit unintended perturbations. The layer is trained via uncertainty-guided bi-level optimization.
The approach is evaluated across vision and language benchmarks demonstrating improvements in group robustness and OOD accuracy.

Reviewers appreciated the clarity of the presentation and the motivations behind the development of the method. The empirical evaluation considering diverse vision and NLP benchmarks was also praised for demonstrating consistent gains in worst-group accuracy and robustness over strong baselines, although some reviewers noted the need for clarifications on the choice of baselines, and ask for more thorough interpretation of the comparison results to fully appreciate their significance.

Methodologically, the paper was praised for its solid basis in established techniques, and some concerns raised regarding the potential scalability limitation of using bi-level optimization were addressed in the rebuttal and revised version with empirical measurements demonstrating manageable overhead.
Another concern related to methodological choices and their lack of motivation was the chosen training scenario without group annotations, an important aspects since, as reviewers noted, the method demonstrates weaker performance than baselines that make use of group annotations.
This concern was partially addressed in the rebuttal by highlighting the advantage of not needing group annotations which avoid possibly time-consuming annotation campaigns requiring domain expertise, but it is still unclear what domains would benefit from this setting and still does not addresses the fact that methods relying on pseudo-annotations are not predicated on extensive human annotations.

The most significant concerns were raised about the limited novelty of the approach, that leverages standard components presented in previous work (Bayesian last layer, sparsity regularization, bi-level optimization) and integrates them with limited methodological innovation and design decisions that are not always motivated from principled theoretical arguments, such as the choice of sparsity prior and the metric for uncertainty quantification.
The paper was also critiqued for the limited theoretical analysis supporting the proposed method, which is rather seen as a heuristic development over the mentioned existing components. Following up on these criticisms regarding significance and novelty, reviews also noted that the title and claims in the abstract about "turning uncertainty into control" were somewhat overstated, as the method primarily focuses on using uncertainty estimates to guide parameter updates rather than providing direct control over model behavior.

The rebuttals and revision additions provided remarkable improvements in terms of addressing clarification behind some methodological choices and validating them empirically, but were not sufficient to fully address the concerns regarding novelty and significance, ultimately leading to the conclusion that while the method is technically sound and addresses an important problem, the paper needs to be repositioned in order to fully address the concerns about novelty, design motivation and theoretical analysis.

**Reviewer Concerns:**

* Addressed in the rebuttal:
  - motivation for investigating annotation-free setting partially addressed in the rebuttal
  - computational cost and scalability of bi-level optimization addressed with empirical measurements

* Not addressed in the rebuttal:
  - need for clarifications on the choice of baselines and more thorough interpretation of the comparison results to fully appreciate their significance not fully addressed
  - limited novelty, arbitrary design choices that aren't being motivated in a principled manner regarding sparsity prior, the metric for uncertainty quantification

**Reviewer Scores:**

| Reviewer | initial score | predicted final score |
|---:|---:|---:|
| g921 | 4 | 4 |
| 3fkM | 2 | 2 |
| 8Xr1 | 6 | 6 |
| jkMj | 4 | 6 |
| RT8U | 4 | 4 |

---

### Decision · Program_Chairs · 2026-01-26

Reject